## LETTERS

# Inflammation triggers ILC3 patrolling of the intestinal barrier

Angélique Jarade [1], Zacarias Garcia[2], Solenne Marie[1], Abdi Demera[3], Immo Prinz [3], Philippe Bousso [2], James P. Di Santo [1] and Nicolas Serafini [1]✉

An orchestrated cellular network, including adaptive lymphocytes and group 3 innate lymphoid cells (ILC3s), maintains intestinal barrier integrity and homeostasis. T cells can monitor environmental insults through constitutive circulation, scanning tissues and forming immunological contacts, a process named immunosurveillance. In contrast, the dynamics of intestinal ILC3s are unknown. Using intravital imaging, we observed that villus ILC3s were largely immotile at steady state but acquired migratory 'patrolling' attributes and enhanced cytokine expression in response to inflammation. We showed that T cells, the chemokine CCL25 and bacterial ligands regulated intestinal ILC3 behavior and that loss of patrolling behavior by interleukin-22 (IL-22)-producing ILC3s altered the intestinal barrier through increased epithelial cell death. Collectively, we identified notable differences between the behavior of ILC3s and T cells, with a prominent adaptation of intestinal ILC3s toward mucosal immunosurveillance after inflammation.

To support intestinal barrier function, both the innate and adaptive immune systems produce IL-22, which orchestrates an immune–epithelial cross talk[1]. An effective IL-22-dependent epithelial response involves the production of antimicrobial peptides that not only contain commensal communities but also restrict pathogenic infections[2,3]. In the intestine, ILC3s represent an early dominant source of IL-22, which is critical for coordinating barrier maintenance at steady state and during bacterial infection[4,5]. While ILC3 activation signals are characterized[4–7], the spatiotemporal regulation of intestinal ILC3 responses is poorly understood. While ILC3 localization should allow rapid responses to environmental and pathogenic signals[8], little is known about ILC3 intratissue dynamics. In particular, whether ILC3s migrate and adapt their cellular behavior in response to environmental cues is to be characterized. In this study, we show that ILC3s in the intestinal villus are largely immotile under steady-state conditions. After activation, these cells acquire a patrolling behavior and IL-22 production, which contribute to maintaining the intestinal barrier integrity. Thus, our data reveal a prominent tissue adaptation of ILC3s to environmental signals, promoting protective immune responses in the gut.

As ILC3s express the transcription factor RORγt and produce IL-22 (encoded by *Rorc* and *Il22*; Extended Data Fig. 1a), we applied intravital multiphoton imaging in *Rorc*^GFP^*Il22*^TdT^ reporter mice to identify *Rorc*^GFP+^ ILC3s and *Rorc*^GFP+^*Il22*^TdT+^ ILC3s *in vivo* (Extended Data Fig. 1b)[9]. Because *Rorc*^GFP+^ helper T cells are also labeled using this approach, we created *Rag2*^−/−^*Rorc*^GFP^*Il22*^TdT^ reporter mice to selectively visualize and track *Il22*^TdT+^ ILC3s (Extended Data Fig. 1c). To study ILC3 responses in a lymphocyte-replete setting, we generated mixed bone marrow chimeras in which irradiated wild-type (WT) recipient mice received bone marrow nucleated cells from *Rag2*^−/−^*Rorc*^GFP^*Il22*^TdT^ and WT mice expressing cyan fluorescent protein (CFP) under the control of the actin promoter (*Actb*^ECFP^). Seven weeks later, *Actb*^ECFP+^ lymphocytes, as well as *Rorc*^GFP+^ ILC3s, 54% of which were *Rorc*^GFP+^*Il22*^TdT+^ ILC3s could be detected in the small intestine of the bone marrow (Extended Data Fig. 1d).

Because intestinal ILC3s are distributed in lamina propria villi and crypts[8], we used intravital imaging to study ILC3 populations in the bone marrow chimera from both of these sites. We observed intestinal *Rorc*^GFP+^ ILC3s and a larger population of *Actb*^ECFP+^ lymphocytes within the intestinal villi (Fig. 1a and Supplementary Video 1). *Actb*^ECFP+^ lymphocytes exhibited diverse in vivo dynamics, ranging from immobility to rapid migration (Fig. 1b,c), which likely reflected heterogeneity within the *Actb*^ECFP+^ population, containing among other things B, T and myeloid cells. *Rorc*^GFP+^ ILC3s in the intestinal villi were mostly nonmotile cells that lacked *Il22*^TdT^ expression (Fig. 1a and Supplementary Video 1). *Rorc*^GFP+^ ILC3s displayed low speed mean (<2 μm min⁻¹) and were largely arrested (arrest coefficient approximately 90%) and confined (straightness ratio approximately 0.18) (Fig. 1b,c). In contrast, *Rorc*^GFP+^ ILC3s in intestinal crypts were clustered in isolated lymphoid follicles (ILFs) and abundantly expressed *Il22* transcripts (Fig. 1a and Supplementary Video 2), as reported previously[10]. ILC3s residing in ILFs exhibited even more restricted motility compared to villus ILC3s (Fig. 1b,c). These observations identified specific features of compartmentalized intestinal ILC3s with overall limited motility and stronger IL-22 expression in crypts at steady state.

NKp46⁺ ILC3s selectively localize to the intestinal villus lamina propria[8]. To understand the cellular dynamic of this ILC3 subset, we imaged *Ncr1*^GFP^*Il22*^TdT^ reporter mice at steady state. Because *Ncr1*^GFP^ is expressed by both group 1 ILCs and NKp46⁺ ILC3s, we focused our analysis on *Ncr1*^GFP+^*Il22*^TdT+^ cells, which represent *Il22*-expressing NKp46⁺ ILC3s (Fig. 2a,b). The migration of *Ncr1*^GFP+^*Il22*^TdT+^ ILC3s was limited (average speed of approximately 3.4 μm min⁻¹) with most cells (>60%) arrested over the track duration (Fig. 2b and Supplementary Video 3). To characterize the impact of acute inflammation on ILC3 homeostasis, function and behavior, *Ncr1*^GFP^*Il22*^TdT^ mice were injected with bacterial flagellin, which can activate ILC3s indirectly through the stimulation of Toll-like receptor 5⁺ myeloid cells, thereby mimicking bacterial infection-induced inflammation[11,12]. Absolute numbers of gut *Ncr1*^GFP+^*Il22*^TdT+^ ILC3s were increased fivefold 5h post-flagellin stimulation (Fig. 2c and Extended Data Fig. 2b,c); analysis of NKp46^+/−^CD49a^+/−^CCR6^+/−^ ILC3s, which include all ILC3 subsets,

[1]Institut Pasteur, Université Paris Cité, Institut National de la Santé et de la Recherche Médicale U1223, Innate Immunity Unit, Paris, France. [2]Institut Pasteur, Université Paris Cité, Institut National de la Santé et de la Recherche Médicale U1223, Dynamics of Immune Responses Unit, Paris, France. [3]Institute of Immunology, Hanover Medical School, Hanover, Germany. ✉e-mail: nicolas.serafini@pasteur.fr

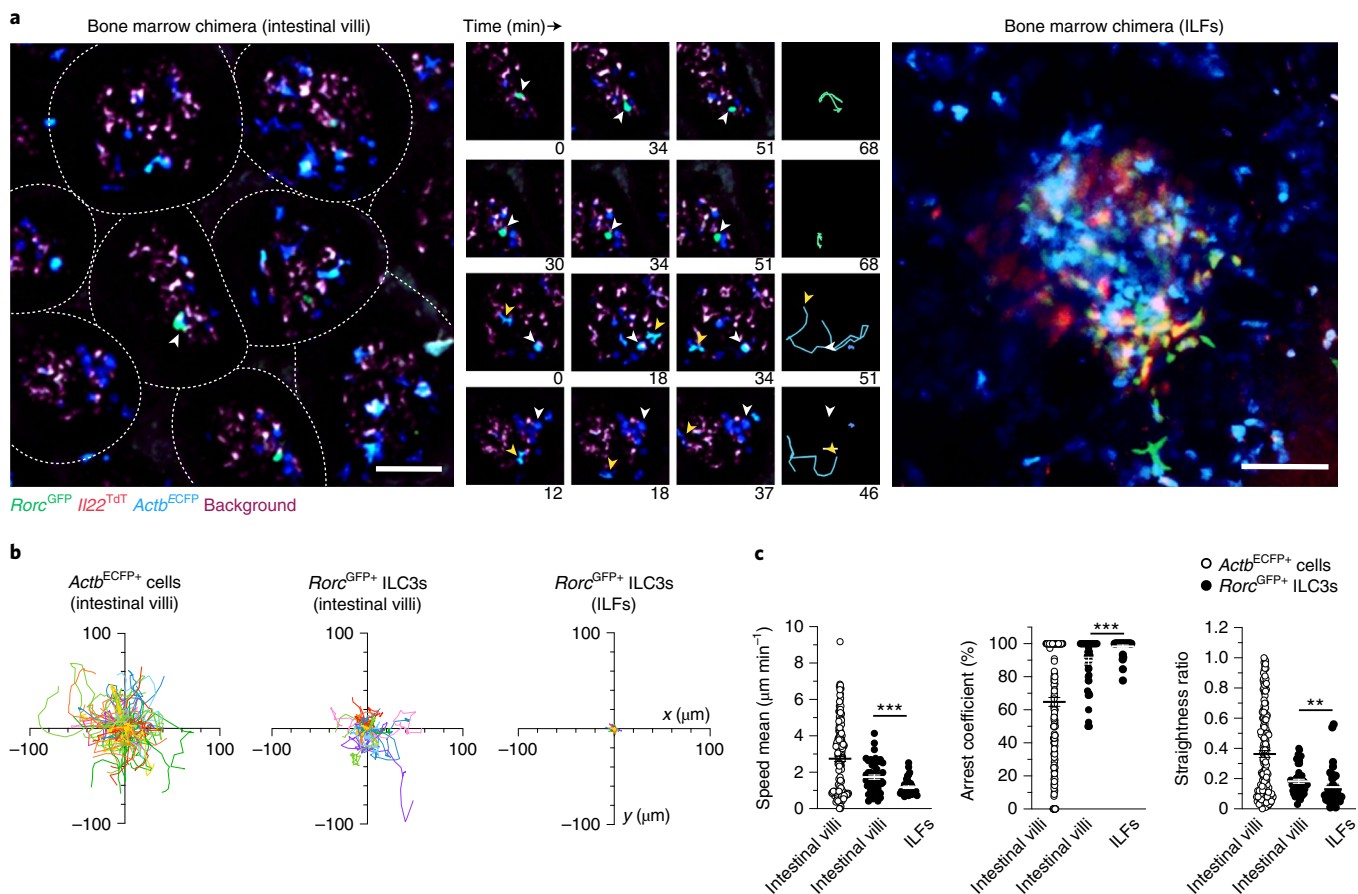

*Rorc*<sup>GFP</sup> *Il22*<sup>TdT</sup> *Actb*<sup>ECFP</sup> Background

**Fig. 1 | Intestinal ILC3s are compartmentalized and poorly motile at steady state. a–c,** Multiphoton microscopy of the small intestine of mixed bone marrow chimeras generated by injection of whole bone marrow CD45.2+ cells from *Actb*<sup>CFP</sup> and *Rag2*<sup>−/−</sup>*Rorc*<sup>GFP</sup>*Il22*<sup>TdT</sup> mice into lethally irradiated congenic CD45.1+ C57BL/6J mice 7 weeks post-transfer. **a,** Representative image (left; scale bar, 50 μm) and time-lapse images (right; scale bar, 15 μm) of ILC3s and CFP+ cells in intestinal villi or ILF of bone marrow chimera. **b,** Individual tracks of intestinal *Actb*<sup>ECFP</sup> cells and *Rorc*<sup>GFP</sup> ILC3s in intestinal villi or ILF. **c,** Mean speed, arrest coefficient and straightness ratio of indicated populations in the intestine. Results in **b,c** are from two (*Actb*<sup>ECFP</sup> cells and *Rorc*<sup>GFP</sup> ILC3s in ILFs) or nine movies (*Rorc*<sup>GFP</sup> ILC3s in intestinal villi) obtained in two independent experiments (n = 170 *Actb*<sup>ECFP</sup> cells; n = 49 *Rorc*<sup>GFP</sup> ILC3ss (intestinal villi); n = 38 *Rorc*<sup>GFP</sup> ILC3s (ILFs)). Each line corresponds to the mean ± s.e.m. of the values obtained; only ILC3s were tested (**$P < 0.002$; ***$P < 0.001$; two-tailed Mann–Whitney U-test; exact P values are provided in the source data).

showed higher frequencies and absolute numbers of IL-22 and *Il22*<sup>TdT</sup>-expressing ILC3s compared to PBS-treated mice (Extended Data Fig. 2c,d). Intravital imaging indicated a marked eightfold increase of *Ncr1*<sup>GFP+</sup>*Il22*<sup>TdT+</sup> ILC3s in intestinal villi, which also displayed increased motility in mice treated with flagellin challenge compared to PBS-treated mice (Fig. 2d, Supplementary Video 4 and Extended Data Fig. 2e,f). *Ncr1*<sup>GFP+</sup>*Il22*<sup>TdT+</sup> ILC3s in flagellin-treated mice migrated within the intestinal villi with enhanced velocity compared to steady-state *Ncr1*<sup>GFP+</sup>*Il22*<sup>TdT+</sup> ILC3s (Fig. 2d–f). Moreover, activated *Ncr1*<sup>GFP+</sup>*Il22*<sup>TdT+</sup> ILC3s exhibited unique movements in the intestinal lamina propria 5 h post-flagellin challenge, shifting from 'spot' migration patterns corresponding to restricted motility to 'wavy' migration patterns (Fig. 2g and Extended Data Fig. 2g). As a result, tissue scanning by *Ncr1*<sup>GFP+</sup>*Il22*<sup>TdT+</sup> ILC3s was increased (Fig. 2h). Together, these data indicate that environmental signals, such as acute generalized inflammation caused by systemic bacterial flagellin administration, impacted ILC3 behavior and induced a patrolling function that was associated with enhanced expression of IL-22.

Intestinal ILC3s are ideally positioned to recognize and process a wide range of external and host-derived signals, allowing them to adapt their effector responses[13]. Increased numbers of intestinal ILC3 and production of IL-22 has been reported in *Rag2*<sup>−/−</sup>

mice compared to WT[13,14]. Intravital imaging indicated that large numbers of patrolling ILC3s were present in the intestinal villi of *Rag2*<sup>−/−</sup>*Rorc*<sup>GFP</sup>*Il22*<sup>TdT</sup> mice at steady state (Fig. 3a and Supplementary Video 5). Villus ILC3s in *Rag2*<sup>−/−</sup>*Rorc*<sup>GFP</sup>*Il22*<sup>TdT</sup> mice constantly scanned intestinal tissue and showed low confinement, reduced arrest coefficient and enhanced velocity compared to WT mice (Fig. 3a–c). Consistent with previous findings[13,14], ILC3s in *Rag2*<sup>−/−</sup>*Rorc*<sup>GFP</sup>*Il22*<sup>TdT</sup> mice had high expression of *Il22*<sup>TdT</sup> compared to WT mice (Extended Data Fig. 1); their enhanced patrolling behavior appeared independent of *Il22* expression (Fig. 3c). These results indicate that dynamic behavior of villus ILC3s is strongly upregulated in the absence of adaptive immune cells.

Reciprocal interactions between intestinal T cells and ILC3s have been reported[15]. To test whether T cells could regulate ILC3 dynamics, we transferred *Actb*<sup>ECFP</sup> T cells in *Rag2*<sup>−/−</sup>*Rorc*<sup>GFP</sup>*Il22*<sup>TdT</sup> mice and imaged their impact on intestinal ILC3s (Extended Data Fig. 3). Compared to ILC3s in *Rag2*<sup>−/−</sup>*Rorc*<sup>GFP</sup>*Il22*<sup>TdT</sup> mice, ILC3s in T cell-reconstituted *Rag2*<sup>−/−</sup>*Rorc*<sup>GFP</sup>*Il22*<sup>TdT</sup> mice showed reduced exploration of the intestinal tissue (straightness ratio approximately 0.25), strongly reduced velocity (<2.2 μm min<sup>−1</sup>) and were mainly arrested (arrest coefficient approximately 85%), regardless of *Il22* expression (Fig. 3a–c and Supplementary Video 6). Thus, the presence of T cells appeared to suppress ILC3 patrolling behavior under

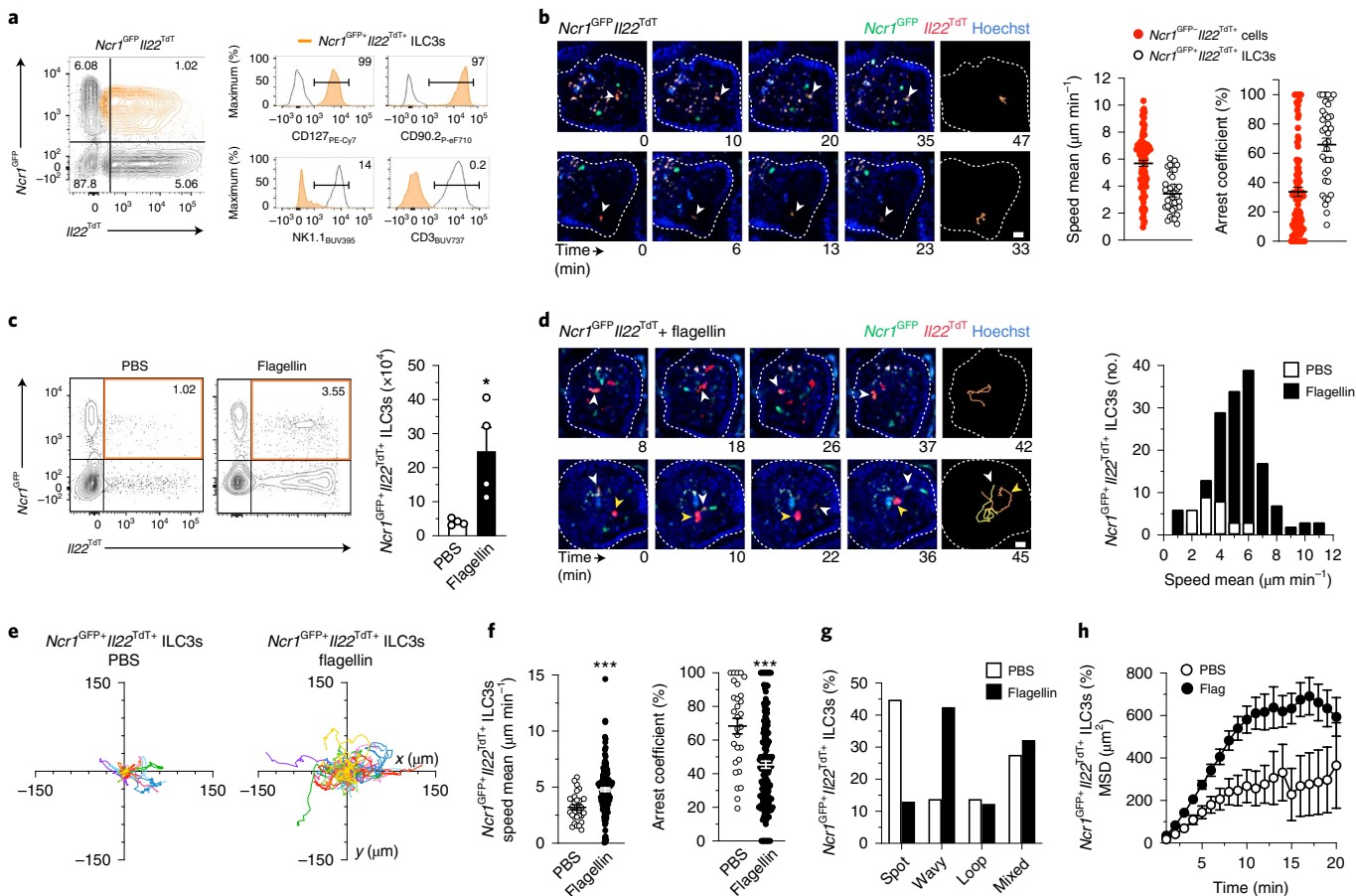

**Fig. 2 | Monitoring of intestinal tissue during inflammation by patrolling ILC3s . a**, Representative flow cytometry analysis for the identification of *Ncr1*GFP+*Il22*TdT+ ILC3s in intestinal CD45+ cells from *Ncr1*GFP*Il22*TdT reporter mice. Data are representative of three independent experiments. **b**, Time-lapse images (scale bar, 15 μm) of *Ncr1*GFP+*Il22*TdT+ cells in intestinal villi. Mean speed and arrest coefficient of *Ncr1*GFP+*Il22*TdT+ ILC3s (*n*=36) compared to *Ncr1*GFP−*Il22*TdT+ cells (*n*=95) **c**, Representative flow cytometry analysis of intestinal *Ncr1*GFP+*Il22*TdT+ ILC3s, pregated on CD45+ cells from one of two independent experiments and absolute numbers of *Ncr1*GFP+*Il22*TdT+ ILC3s (*n*=4 mice per condition) in *Ncr1*GFP*Il22*TdT mice treated or not with flagellin 5 h before. **d**, Time-lapse images (scale bar, 15 μm) of *Ncr1*GFP+*Il22*TdT+ ILC3s in intestinal villi 5 h post-flagellin stimulation as in **c**. Mean speed distribution of intestinal *Ncr1*GFP+*Il22*TdT+ ILC3s, with or without flagellin. **e–h**, Individual tracks (**e**), mean speed and arrest coefficient (**f**), distribution of ILC3 patrolling behaviors (**g**) and MSD (**h**) of *Ncr1*GFP+*Il22*TdT+ ILC3s as in **c** (*n*=29 and 156). Results are from at least nine movies per condition obtained in three independent experiments. Each bar corresponds to the mean ± s.e.m. of the values obtained; only *Ncr1*GFP+*Il22*TdT+ ILC3s were tested (**P* < 0.05; ****P* < 0.001; two-tailed Mann–Whitney *U*-test; exact *P* values are provided in the source data).

steady-state conditions. Because T cells can restrict ILC3 activation in part by limiting the expansion of specific microbial species[16–18], we studied the impact of the microbiota on ILC3 migratory behavior. We treated *Rag2*−/−*Rorc*GFP*Il22*TdT mice for two weeks with antibiotics (including ampicillin, streptomycin and colistin) which strongly deplete microbiota in adult mice[19]. Motility of all ILC3s was decreased after antibiotic treatment compared to water-treated mice (Extended Data Fig. 4). However, ILC3 exploration of the intestinal tissue, based on the straightness ratio, was not changed compared to controls (Extended Data Fig. 4). Hence, while altered microbiota may partially promote ILC3 patrolling in *Rag2*−/− mice, T cells are the main regulators of this process and do not rely on cues from the microbiota.

To assess the balance between T cell-mediated suppression and inflammation-mediated activation of ILC3 patrolling, we challenged T cell-reconstituted *Rag2*−/−*Rorc*GFP*Il22*TdT mice with flagellin (Fig. 3c). In this setting, despite the presence of T cells, both *Il22*TdT− and *Il22*TdT+ intestinal ILC3s exhibited enhanced patrolling behavior compared to PBS-treated mice, with increased velocity and reduced arrest and confinement (Fig. 3c–e and Supplementary Video 7).

Considering that T cells patrol the intestine under physiological and inflammatory conditions, these data suggested that regulation of intestinal T cells and ILC3 patrolling behavior could be linked.

We next investigated the molecular mechanisms driving the ILC3 migration patterns in intestinal villi. Because chemokines play a fundamental role in lymphocyte migration and function[20] and are involved in ILC3 homing and trafficking[8,17,18], we tested whether chemokine–chemokine receptors regulated intestinal ILC3 dynamic behavior. *Ccr6*, *Ccr9* and *Cxcr6* were expressed on intestinal NKp46+ ILC3s and CCR6+ ILC3s[8,18,21]. We further identified the expression of *Ccr7* and *Cxcr4* transcripts and CCR7, CXCR4 and CXCR6 protein expression on both ILC3 subsets—albeit in various amounts—and CCR9 on NKp46+ ILC3s (Fig. 4a,b). Among the relevant chemokines (*Ccl19* and *Ccl21* for CCR7, *Cxcl12* for CXCR4, *Cxcl16* for CXCR6, *Ccl25* for CCR9 and *Ccl20* for CCR6), *Ccl25*, *Cxcl12* and *Cxcl16* were upregulated in whole ileal tissue extracts from *Rag2*−/− mice compared to WT and T cell-reconstituted *Rag2*−/− mice; *Ccl25* was the most upregulated and abundant chemokine (Fig. 4c). To assess the role for CXCL12, CXCL16, CCL25 and CCL21 on ILC3 patrolling in vivo, we blocked their activity

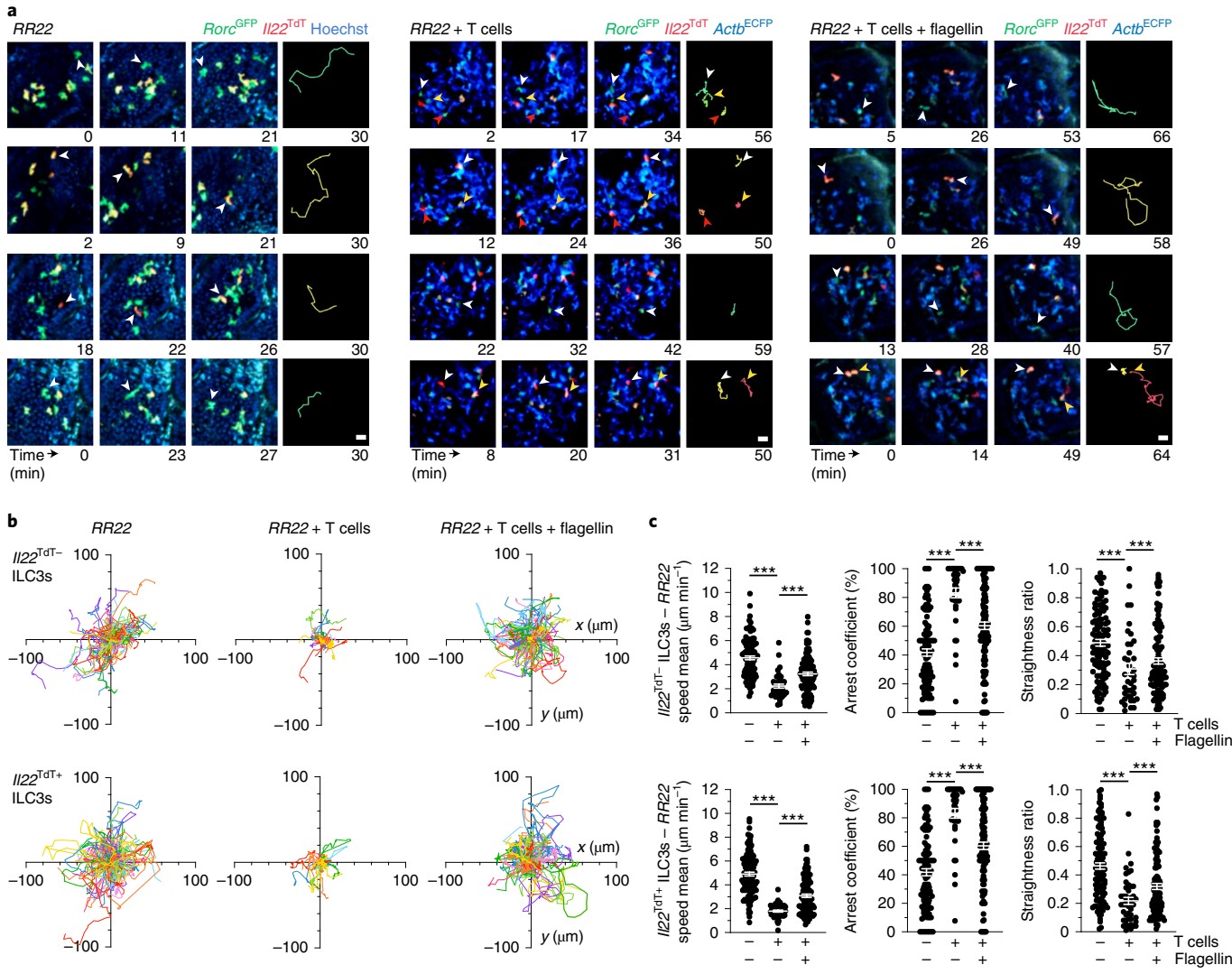

**Fig. 3 | Local environmental stimuli regulate ILC3 patrolling. a**, Representative time-lapse images (scale bar, 15 μm) of ILC3s within intestinal villi of the indicated mice. Data are representative of three independent experiments. **b**, Individual tracks of $Il22^{TdT-}$ or $Il22^{TdT+}$ $Rorc^{GFP+}$ ILC3s in noted mice. **c**, Mean speed, arrest coefficient and straightness ratio of $Il22^{TdT-}$ and $Il22^{TdT+}$ ILC3s as in **a**. Results are from 1–3 movies per condition from 1 of 3 independent experiments ($n = 109$ and $n = 120$ for $Il22^{TdT-}$ and $Il22^{TdT+}Rorc^{GFP+}$ ILC3s in RR22 mice, respectively and $n = 39$ and 120 $Il22^{TdT-}Rorc^{GFP+}$ ILC3s; $n = 45$ and 109 $Il22^{TdT+}Rorc^{GFP+}$ ILC3s for RR22 + $Actb^{CFP+}$ T cell mice without or with flagellin, respectively). Each bar corresponds to the mean ± s.e.m. of the values obtained (***$P < 0.001$; one-way analysis of variance (ANOVA); exact $P$ values are provided in the source data).

with neutralizing antibodies (Fig. 4d). Imaging of intestinal ILC3s in $Rag2^{-/-}Rorc^{GFP}Il22^{TdT}$ mice before and immediately after injection with antibodies against CXCL12, CXCL16, CCL25 and CCL21 indicated that the velocity of intestinal ILC3s strongly decreased 20 min after neutralization (Fig. 4e, Extended Data Fig. 5a–c and Supplementary Videos 8 and 9). Chemokine blockade impaired ILC3 patrolling and tissue scanning, which was reflected by a reduction in villus ILC3 trajectories, average speed and straightness ratios compared to isotype-injected mice (Fig. 4f,g). Patrolling ILC3s lost motility with a high arrest coefficient (over 70%) (Fig. 4f,g). The effects of combined CXCL12, CXCL16, CCL25 and CCL21 blockade were observed within the first hour postinjection and intensified with time (30–60 min later; Fig. 4g).

Next, we analyzed the specific role of CCL25 in controlling the patrolling behavior of ILC3s. Administration of CCL25-specific antibodies to $Rag2^{-/-}Rorc^{GFP}Il22^{TdT}$ mice reduced villus ILC3 patrolling compared to isotype-treated mice, with a progressive decrease in ILC3 speed 30 min after injection (Fig. 5a,b,

Extended Data Fig. 5d–f and Supplementary Videos 10 and 11). CCL25 blockade limited ILC3 migration, with reduced trajectories, speed mean (2 μm min⁻¹), augmented arrest coefficient (approximately 85%) and confinement (approximately 0.3) compared to isotype-injected mice (Fig. 5c,d). Combined neutralization of CXCL12, CXCL16 and CCL21 was less effective than CCL25 blockade in inhibiting ILC3 motility, with modest reduction of ILC3 velocity and time arrested at 1 and 2 h postinjection (Extended Data Fig. 6a), suggesting that CCR9-CCL25 signaling was a critical regulator of ILC3 patrolling in the intestine. To evaluate whether CCL25 controlled inflammation-induced intratissular migration of ILC3s in WT mice, we imaged intestinal $Ncr1^{GFP+}Il22^{TdT+}$ ILC3s in flagellin-treated mice before, 1 h and 2 h after injection with CCL25-blocking antibodies. Neutralization of CCL25 reduced $Ncr1^{GFP+}Il22^{TdT+}$ ILC3 patrolling trajectories and dynamics (Extended Data Fig. 6b), suggesting that CCL25 was an important signal involved in the ILC3 migratory behavior in response to generalized inflammation.

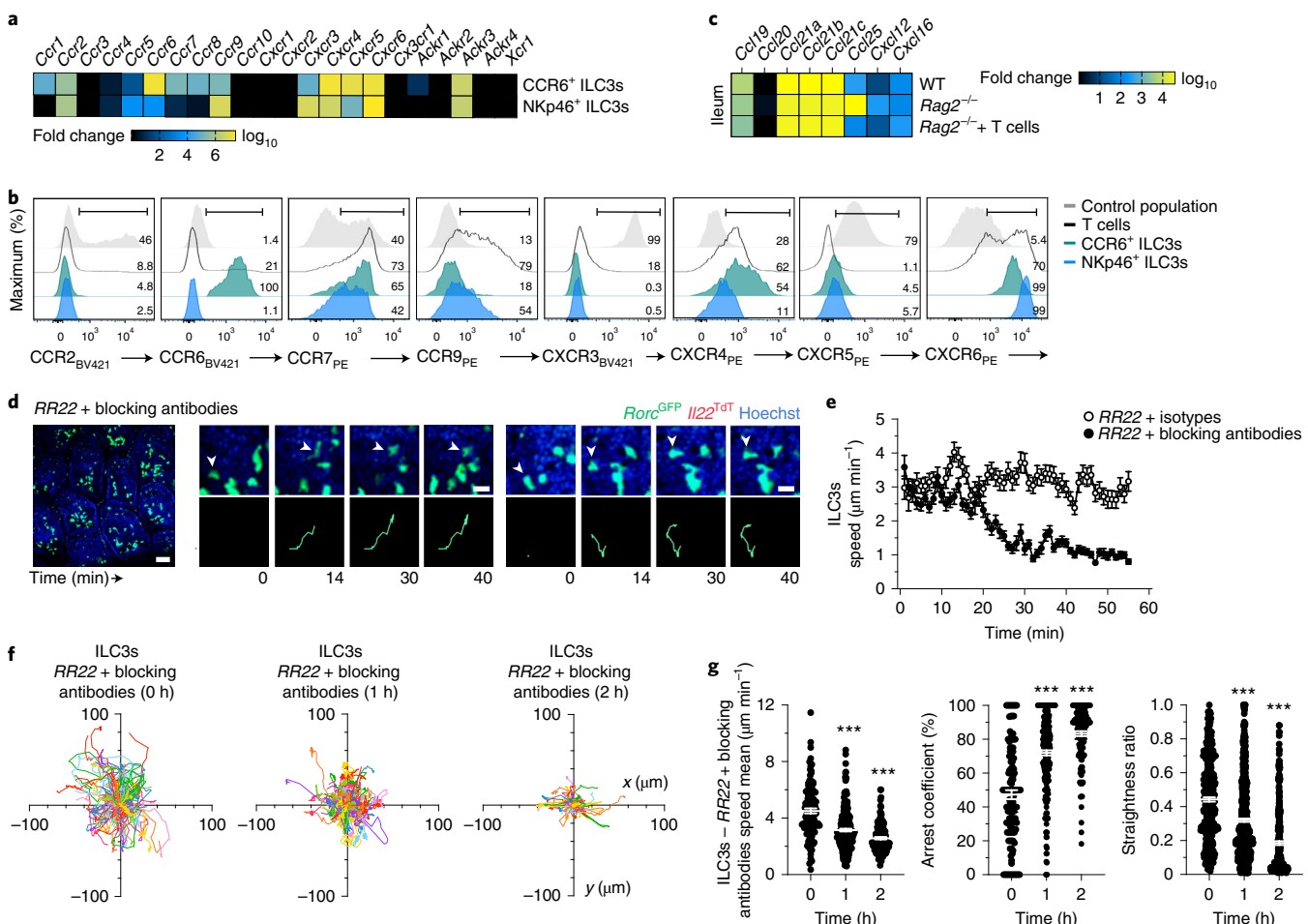

**Fig. 4 | Chemokines are involved in the control of ILC3 motility. a**, Heatmap of relative expression of chemokine receptors in sorted NKp46[+] ILC3s ($n = 6$) and CCR6[+] ILC3s ($n = 6$) from WT mice detected by bulk multiplex Biomark assay. **b**, Representative flow cytometry analysis of chemokine receptors profiles of intestinal NKp46[+] ILC3s, CCR6[+] ILC3s and T cells from one of three independent experiments in WT mice. A negative or positive population is shown as a control for each chemokine receptor. **c**, Heatmap of relative expression of chemokines in whole ileum of WT ($n = 10$), $Rag2^{-/-}$ ($n = 10$) and $Rag2^{-/-}$ adoptively transferred with T cells ($n = 8$) detected by bulk multiplex Biomark assay. **d–g**, Intravital imaging of intestinal ILC3s in RR22 mice with combination of isotype controls (mouse IgG1, rat IgG2a and rat IgG2b) or blocking monoclonal antibodies (anti-CXCL12, anti-CXCL16, anti-CCL21 and anti-CCL25). Data are representative of three independent experiments. Representative image (left; scale bar, 50 μm), time-lapse images (**d**) (right; scale bar, 15 μm), speed over time (**e**) ($n = 216$ and 248), individual tracks (**f**), mean speed, arrest coefficient and straightness ratio (**g**) of intestinal ILC3s as in **d** at the indicated time points. Results in **f,g** are from 3 movies per condition obtained in 3 independent experiments ($n = 364$, 0 h; $n = 617$, 1h; $n = 310$, 2 h). Each bar corresponds to the mean ± s.e.m. of the values obtained (***$P < 0.001$; one-way ANOVA; exact $P$ values are provided in the source data).

Since a subset of intestinal T cells express CCR9 (refs. [22]), we assessed whether T cells may regulate ILC3 patrolling by competing for CCL25 that drive ILC3 migration. To investigate the role of CCR9[+] T cells in ILC3 motility, we adoptively transferred T cells from WT or $Ccr9^{-/-}$ mice into $Rag2^{-/-}Rorc^{GFP}Il22^{TdT}$ recipients. While villus ILC3 patrolling was ablated in mice transferred with WT T cells, ILC3s exhibited a patrolling behavior in mice that received $Ccr9^{-/-}$ T cells (Fig. 5e, Extended Data Fig. 7 and Supplementary Video 12), suggesting that T cell regulation of ILC3 patrolling occurred, at least in part, through the regulation of CCL25 availability by CCR9[+] T cells. To assess whether T cells regulated ILC3 patrolling through buffering CCL25, we treated $Rag2^{-/-}Rorc^{GFP}Il22^{TdT}$ mice that had been adoptively transferred with $Ccr9^{-/-}$ T cells with CCL25 blocking antibodies. One hour post-CCL25 neutralization, ILC3 patrolling was reduced, with ILC3s arresting and displaying reduced speed mean as well as straightness ratio compared to isotype-treated mice (Fig. 5f and Supplementary Video 13). These findings indicated that regulation of CCL25 availability by CCR9[+] T cells represented one mechanism by which T cells regulate ILC3 patrolling.

To investigate whether ILC3 patrolling had an impact on the intestinal barrier, we treated $Rag2^{-/-}$ mice with CCL25 blocking antibodies and used oral gavage of fluorescein isothiocyanate (FITC)-dextran to measure intestinal permeability. $Rag2^{-/-}$ mice treated with CCL25 blocking antibodies exhibited a significant increase in FITC-dextran in the serum compared to isotype-treated mice (Fig. 5g). Because IL-22–IL-22 receptor interactions and ILC3s are responsible for homeostatic signaling and activation of intestinal epithelial cells (IECs)[16,23,24], we investigated the influence of ILC3 patrolling on mechanisms involved in preserving barrier integrity, including epithelial expression of tight junction proteins and cell death. Expression of tight junctions and adhesion proteins was similar in isotype and CCL25 antibody-treated $Rag2^{-/-}$ mice (Fig. 5h). However villus IEC death—as detected by immunofluorescence staining of cleaved caspase-3—was increased in $Rag2^{-/-}$ mice after CCL25 neutralization (Fig. 5i), suggesting that patrolling ILC3s have a role in epithelial homeostasis. To investigate whether this protection is mediated by IL-22, we quantified IEC death in isotype and CCL25 antibody-treated

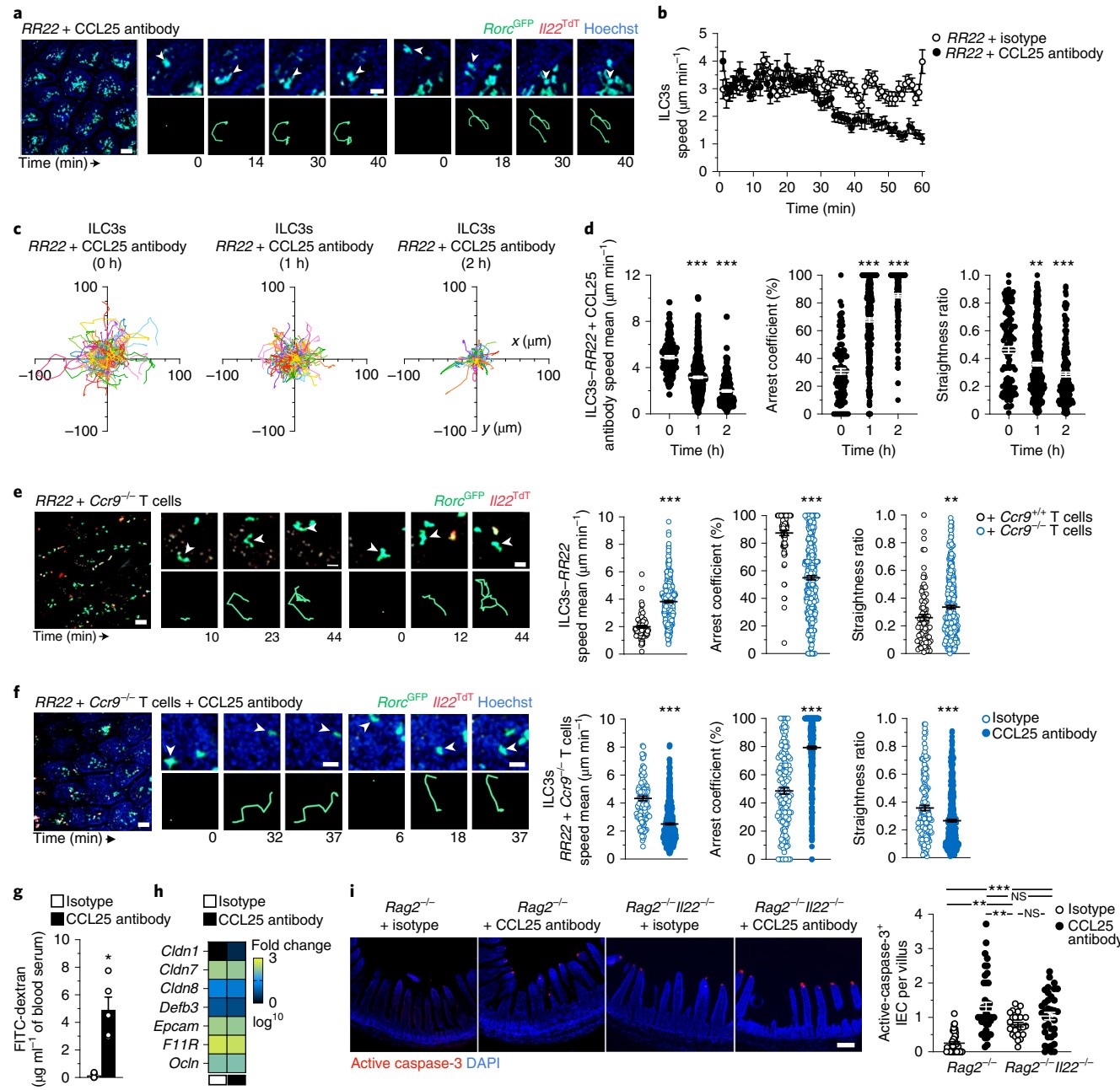

**Fig. 5 | IEC death is prevented by ILC3 patrolling, which is regulated by CCL25 and T cells. a–f,** Intravital imaging of intestinal ILC3s in *RR22* mice injected with isotype or CCL25 antibody (**a–d,f**), with or without *Ccr9*[+/+] (**e**) or *Ccr9*[−/−] T cells (**e,f**) showing representative images (left; scale bar, 50 μm), time-lapse images (right; scale bar, 15 μm) of villus ILC3s (**a,e,f**). Speed over time of intestinal ILC3s in *RR22*, after either isotype or CCL25 antibody injection at time 0 (**b**), individual tracks (**c**), mean speed, arrest coefficient and straightness ratio (**d–f**) of intestinal ILC3s at 1 h (**d–f**) and 2 h (**d**) or 1 h with isotype or CCL25 antibody (**b–d**), *Ccr9*[+/+] or *Ccr9*[−/−] T cells (**e**), isotype or CCL25 antibody + *Ccr9*[−/−] T cells (**f**). Data are representative of three independent experiments (**a,c,f**). Results in **b–f** are from at least 2 movies per condition obtained in 2 independent experiments (**b–d**: *n* = 121, 0 h; *n* = 400, 1 h; *n* = 179, 2 h; **e**: *n* = 84, RR22+ *Ccr9*[+/+] T cells and *n* = 338, *RR22*+ *Ccr9*[−/−] T cells; **f**: *n* = 152, isotype and *n* = 516, CCL25 antibody). **g**, Intestinal permeability assay in *Rag2*[−/−] mice injected intravenously with either isotype or CCL25 antibody 18 h and 4 h before analysis. **h**, Heatmap of tight junctions and adhesion molecule expression in whole ileum (*n* = 5) in *Rag2*[−/−] mice as in **g**, detected by Biomark assay. **i**, Representative immunofluorescence analysis (ileum; scale bar, 50 μm) and absolute numbers of active caspase-3[+] IECs in isotype or CCL25 antibody-treated *Rag2*[−/−] and *Rag2*[−/−]*Il22*[−/−] mice. Results in **i** are from two independent experiments (*n* = 4 mice *Rag2*[−/−] and *n* = 43 out of 48 fields for each condition; *n* = 3 *Rag2*[−/−]*Il22*[−/−] mice and *n* = 27 out of 37 fields for each condition). Each bar corresponds to the mean ± s.e.m. of the values obtained (NS, not significant; *P < 0.05; **P < 0.01; ***P < 0.001; one-way ANOVA in **d–f** and two-tailed Mann–Whitney *U*-test in **g–i**; exact *P* values are provided in the source data).

*Rag2*[−/−]*Il22*[−/−] mice. Ileal IEC death was higher in *Rag2*[−/−]*Il22*[−/−] mice compared to *Rag2*[−/−] mice and at levels comparable to that seen in *Rag2*[−/−] mice treated with CCL25 blocking antibodies (Fig. 5i); CCL25 antibody neutralization in *Rag2*[−/−]*Il22*[−/−] mice did not further increase IEC death compared to isotype-treated *Rag2*[−/−]*Il22*[−/−] mice (Fig. 5i). These observations suggested that

patrolling ILC3s delivered homeostatic signals, including IL-22, that maintained the intestinal barrier (Extended Data Fig. 8).

In this study, we provided an in-depth characterization of intestinal ILC3 behavior. We showed that ILC3s were compartmentalized in the intestine and largely immotile under basal conditions, in both the intestinal villus and crypts. We showed that environmental signals provided tissue-resident ILC3s with new migratory attributes. On inflammation, villus ILC3s rapidly increased their motility to become patrolling cells, resulting in enhanced tissue scanning. While most immune cells behaved as 'explorers' that exhibit robust cellular migration under basal conditions and reduced motility after activation[25], our study detected a unique behavior for ILC3 and suggested a new 'gatekeeper' model, in which tissue protection operates through distinct T and ILC3 migration patterns. Our study showed an important role of T cells in regulating intratissular ILC3 migration through competition for the chemokine CCL25 and identified the CCL25–CCR9 axis as one of the signals controlling patrolling ILC3s. Thus, we propose that microenvironmental signals promote tissue adaptation of intestinal ILC3 responses, including patrolling behavior and IL-22 production, which contributes to maintaining the balance between regulatory and immune protective ILC3 functions. As such, we suggest that patrolling ILC3s help to diffuse IL-22 within the villi to prevent epithelial cell death and support intestinal barrier integrity. Dysregulated ILC3 responses have been associated with the development of intestinal pathology, such as inflammatory bowel diseases[26]. Therefore, a better understanding of ILC3 behavior and associated regulatory mechanisms may lead to future therapeutic approaches for these debilitating diseases.

## Online content

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

## Methods

**Mice.** All mice used were on a C57BL/6 background and bred in dedicated animal facilities of the Institut Pasteur. $Actb^{ECFP}$, $Ccr9^{-/-}$, $Rorc^{GFP}$, $Il22^{TdT}$ and $Ncr1^{GFP}$ mice were provided by P. Bousso, I. Prinz, G. Eberl, S. Durum and O. Mandelboim, respectively. C57BL/6 and C57BL/6 Ly5.1 mice were purchased from the Charles River Laboratories. All experiments were performed on 8–14-week-old male and female animals, except for experiments on bone marrow chimeric mice which involved 16-week-old males at the time of the analysis. All experiments involving mice were performed according to guidelines issued by the Institut Pasteur Ethics Committee and were approved by the French Ministry of Research (project nos. DHA170001, CETEA 2013-033 and CETEA 17500).

**Adoptive transfer of T cells.** Cells from the spleen and lymph nodes of $Actb^{ECFP}$ reporter or $Ccr9^{-/-}$ mice were isolated by being passed through a steel wire mesh, filtered on a 40-µm cell strainer and after red blood cells lysis using ACK buffer. To enrich for T cells, single-cell suspensions were first incubated with biotin-conjugated anti-B220 (clone RA3-6B2; eBioscience), anti-CD19 (clone 1D3; eBioscience), anti-CD11b (clone M1/70; BD Biosciences), anti-CD11c (clone N418; eBioscience), anti-Gr1 (clone RB6-8C5; eBioscience); anti-NK1.1 (clone PK136; BD Biosciences) and anti-Ter119 (clone TER-119; BD Biosciences) antibodies followed by antibiotin microbeads (Miltenyi Biotec) and negatively selected by magnetic cell separation with magnetic-activated cell sorting technology (Miltenyi Biotec); $5 \times 10^6$ enriched T cells were intravenously transferred to $Rag2^{-/-}Rorc^{GFP}Il2 2^{TdT}$ ($RR22$) mice; recipient mice were analyzed at day 14 post-transfer.

**Generation of bone marrow chimeric mice.** Cells from the bone marrow of $Actb^{ECFP}$ and $RR22$ mice were isolated by crushing the bones with a mortar and pestle, being filtered on a 40-µm cell strainer and after red blood cells lysis using the ACK buffer. Bone marrow cells were mixed at a 1:19 ratio—$0.5 \times 10^6$ and $10 \times 10^6$ cells from $Actb^{ECFP}$ and $RR22$ mice, respectively. Mixed bone marrow cells were transferred intravenously into lethally irradiated (9 Gy X-ray irradiation 1 d before injection) congenic C57BL/6J mice. Bone marrow chimeric mice were analyzed seven weeks later for reconstitution (donor cells in the small intestine lamina propria) and imaged by multiphoton microscopy.

**Isolation of intestinal cells.** Small intestine was collected from euthanized mice and placed into cold complete medium: Roswell Park Memorial Institute (RPMI) 1640 GlutaMAX (Gibco) supplemented with 5% fetal calf serum (FCS) (Eurobio) and 10 mM HEPES (Sigma-Aldrich). The mesenteric adipose tissue and Peyer's patches were first pulled out before cutting the small intestine longitudinally and removing feces. Intestinal tissue was washed in PBS (Gibco) to eliminate mucus, cut into 1–2-cm pieces and IECs were eliminated by shaking incubation in complete medium containing 5 mM EDTA (Invitrogen) for 20 min at 37 °C. Subsequently, intestinal tissue was minced and incubated twice in a digestion solution of complete medium containing Collagenase VII (0.5 mg ml$^{-1}$; Sigma-Aldrich) for 15 min at 37 °C in a shaking incubator to isolate the lamina propria lymphocytes. Lamina propria lymphocytes were filtered through a 40-µm cell strainer and kept in complete medium for downstream analysis.

**Flow cytometry.** Cells were first blocked with FcR Blocking Reagent (Miltenyi Biotec) and stained with Flexible Viability Dye (eFluor 506 or 780; eBioscience) for 15 min, followed by 30 min of surface antibody staining on ice except for chemokine receptors (at 37 °C for 30 min then at 4 °C for 15 min). Cells were generally fixed in 4% paraformaldehyde (PFA) (Sigma-Aldrich). Only for experiments involving intranuclear transcription factor staining, cells were fixed, permeabilized and stained using Foxp3/Transcription Factor Staining Buffer Kit (eBioscience). Intranuclear and cell surface staining were performed with the following antibodies from BD Biosciences, eBioscience and BioLegend: anti-CCR2 (clone SA203G11); anti-CCR6 (clone 140706); anti-CCR7 (clone 4B12); anti-CCR9 (clone CW-1.2); anti-CD11b (clone M1/70); anti-CD19 (clone 1D3); anti-CD127 (clone A7R34); anti-CD3 (clone 145-2C11); anti-CD3 (clone 500A2 or 17A2); CD44 (clone IM7); anti-CD45 (clone 30-F11); anti-CD45.1 (clone A20); anti-CD45.2 (clone 104); anti-CD5 (clone 53-7.3); CD62L (clone MEL-14); anti-CD8a (clone 53-6.7); anti-CD4 (clone GK1.5); anti-CD90.2 (clone 30-H12); anti-CD90.2 (clone 53-2.1); anti-CXCR3 (clone CXCR3-173); anti-CXCR4 (clone QA16A08); anti-CXCR5 (clone L138D7); anti-CXCR6 (clone SA051D1); Foxp3 (clone FJK-16s); Gata-3 (clone TWAJ); anti-KLRG1 (clone 2F1); anti-NK1.1 (clone PK136); anti-NKp46 (clone 29A1.4); anti-TCRβ (clone Q31-378); and anti-TCRβ (clone H57-597). All samples were acquired on a custom-configured LSR Fortessa or sorted using a FACSAria III (BD Biosciences). To sort the ILC3 subsets, we applied the following gating strategies: live CD45.2$^+$CD3$^-$NK1.1$^-$KLRG1$^-$CD90.2$^+$CD127$^+$ and subgated on NKp46$^+$ or CCR6$^+$ cells. Data were analyzed with the FlowJo 10 software (FlowJo LLC).

**Intravital two-photon imaging.** Mice were first anesthetized with a mixture of ketamine (62.5 mg kg$^{-1}$), xylazine (12.5 mg kg$^{-1}$) and acepromazine (3.1 mg kg$^{-1}$) and the abdomen hair was shaved. The abdominal skin and wall musculature were incised along the linea alba to expose the intestine. A loop of the terminal ileum (2-cm upstream the cecum) was exposed and cut open on the opposite side of the mesentery using an electrical cautery. Feces were gently removed using PBS

and mice were placed on a heated (37 °C) steel plate. Mice were immobilized by placing polyvinyl siloxane-based paste (3M) on both sides of the abdomen and a SuperFrost Plus slide (Menzel Gläser; VWR) was placed across the paste with a PBS-soaked tissue paper (Kimtech; Kimberly-Clark Corporation) on top. The intestinal tissue was positioned on the slide, immobilized with a coverslip (Menzel Gläser; VWR) on top held by paper clips and covered with Supragel ultrasound gel (LCH). During imaging, mice were supplied with oxygen and their temperature was controlled and maintained at 37 °C using a heating pad, heating cover and objective heater for the tissue. When indicated, intestine and blood vessels labeling was performed by intravenous injection of Hoechst 33342 (40 µl of 10 mM solution, prepared in PBS; Thermo Fisher Scientific) and Evans Blue (25 µl of 2 µg µl$^{-1}$ solution, prepared in PBS; Sigma-Aldrich), respectively. Two-photon imaging was performed with an upright microscope FVMPE-RS (Olympus Lifescience) and a ×25, 1.05-numerical aperture, water-dipping objective (Olympus Lifescience). Excitation was provided by an InSight DeepSee dual laser (Spectra-Physics) tuned at 920. The following filters were used for fluorescence detection: CFP or Hoechst (483/32), GFP (520/35), TdT (593/40) and background (624/40). To create time-lapse sequences, we typically scanned a 40–60-µm-thick volume of tissue at 7-µm Z-steps and 60-s intervals.

**Image analysis.** Movies were processed with the Imaris 7.4.2 (Bitplane) or Fiji 2 (ImageJ2) software and analyzed as two-dimensional (2D) projections of three-dimensional (3D) data to avoid wrong tracking related to motion (intestinal villi or diaphragmatic movement). As such, the represented speeds of motile cells are likely to be slightly underestimated. When necessary, drifting correction was also applied using the 'Correct Drift' function in Imaris to minimize XY tissue drift. DiPer[27] was used to calculate mean square displacement (MSD) and tracking behaviors. Movies and figures based on two-photon microscopy are shown as 2D projections of 3D data. For optimal contrast rendering, background was pseudocolored in magenta in some images. The arrest coefficient was defined as the percentage of time where instantaneous velocity was <3 µm min$^{-1}$. The straightness ratio was defined as the ratio track displacement/track path length that reflects directionality.

**In vivo treatments.** To induce intestinal inflammation, mice were injected intravenously with purified flagellin from *Salmonella typhimurium* (FLA-ST Ultrapure, 5 µg; Invivogen) 5 h before analysis. For microbiota depletion, mice were orally treated for 2 weeks with a combination of antibiotics, administered in their drinking water, including ampicillin (1 mg ml$^{-1}$), streptomycin (5 mg ml$^{-1}$), colistin (1 mg ml$^{-1}$) and sucrose (25 mg l$^{-1}$), all from Sigma-Aldrich. For chemokine neutralization, mice were injected intravenously during imaging with a mixture of Hoechst 33342 (see above) to control intravenous injection and the following monoclonal blocking antibodies (50 µg): anti-CXCL12 (catalog no. MAB310); anti-CXCL16 (catalog no. MAB503); anti-CCL21 (catalog no. MAB4572) and anti-CCL25 (catalog no. MAB481), all from R&D Systems. Control mice were injected intravenously in accordance with the following isotype controls (50 µg): mouse IgG1 (catalog no. MAB002); rat IgG2a (catalog no. MAB006); and rat IgG2b (catalog no. MAB0061), all from R&D Systems.

**Immunofluorescence staining and confocal imaging.** The ileal portion of the small intestine was cut longitudinally, washed in PBS and prepared using the Swiss-rolling technique. Intestinal rolls were fixed overnight in 4% PFA followed by dehydration in 30% sucrose (Sigma-Aldrich) before embedding in Tissue-Tek OCT compound (Sakura Finetek). Samples were frozen in an isopentane bath cooled with liquid nitrogen and stocked at −80 °C; 8-µm sections were cut on a CM3050 S cryostat (Leica Biosystems) and adhered to SuperFrost Plus slides. Frozen sections were first hydrated with PBS-TS (PBS supplemented with 0.1% Triton X-100, 1% FCS, 1% bovine serum albumin (Sigma-Aldrich) and filtered) and blocked for 1 h at room temperature with PBS-TS 10% FCS. Slides were then incubated overnight at 4 °C with the following antibodies diluted in PBS-TS: anti-red fluorescent protein (Rockland); phalloidin (Invitrogen); anti-CD3 (clone 500A2; BD Biosciences); and anti-active caspase-3 (clone C92-605; BD Biosciences). The next day, slides were washed, incubated for 1 h at room temperature with secondary conjugated antibodies, washed again and incubated for 2 min at room temperature with 4,6-diamidino-2-phenylindole (DAPI) (for nuclei, 1 µg ml$^{-1}$; Sigma-Aldrich). After staining, slides were mounted with Fluoromount-G (SouthernBiotech) and examined on an Axio Imager Z.2 microscope (ZEISS).

**In vivo intestinal permeability assay.** Mice received FITC-dextran (4 kDa, 12 mg g$^{-1}$ of body weight, Sigma-Aldrich) by oral gavage. One hour later, sera were collected and analyzed for fluorescence intensity using the FLUOstar OPTIMA (BMG Labtech).

**Bulk RNA isolation and multiplex quantitative PCR.** For ILC3, $3.3–7.4 \times 10^3$ and $2.2–4.5 \times 10^3$ NKp46$^+$ and CCR6$^+$ ILC3 bulks were sorted directly into RLT buffer. For whole ileal tissue, dissected ileum was snap-frozen in liquid nitrogen, crushed with a mortar and pestle and homogenized into RLT buffer. Samples were stored at −80 °C until messenger RNA purification using RNeasy Micro and Minikit with a DNase digestion step using the RNase-Free DNase Set (QIAGEN) for ILC3s and ilea, respectively. mRNA quantity, quality and integrity were checked on a Bioanalyzer system (Agilent) or on a Nanodrop

Spectrophotometer (Thermo Fisher Scientific) for ILC3s and ilea, respectively. To synthesize complementary DNA, PCR with reverse transcription was performed using the Transcriptor High Fidelity cDNA Synthesis Kit (Roche). From the cDNA, we followed the protocol 'Fast Gene Expression Analysis Using Evagreen on the Biomark' from Fluidigm. Briefly, preamplified cDNA was obtained after a preamplification step with Delta Genes Assays (Fluidigm) using the TaqMan PreAmp Master Mix (Applied Biosystems) and a subsequent purification step using exonuclease I (New England Biolabs) and was then diluted 1:5 in Tris EDTA buffer (Invitrogen). The sample mix was prepared as follows: diluted preamplified cDNA (3.6 μl), DNA Binding Dye Sample Loading Reagent (4 μl; Fluidigm) and SsoFast EvaGreen Supermix with low ROX (0.4 μl; Bio-Rad Laboratories). The assay mix was prepared as follows: primers (4 μl, 10 μM; Fluidigm) and assay loading reagent (4 μl; Fluidigm). A 96.96 Dynamic Array Integrated Fluidic Circuit (Fluidigm) was primed with control line fluid and the chip was loaded with assays and samples using an HX Integrated Fluidic Circuit controller (Fluidigm). The experiments were run on a Biomark HD (Fluidigm) for amplification and detection (70 °C, 2,400 s; 60 °C, 30 s; 95 °C, 60 s; 96 °C 5 s, 60 °C, 20 s) 30 cycles; melting curve: 60 °C, 3 s; 60–95 °C 1 °C 3 s$^{-1}$). Real-time PCR analysis 4.5.2 software (Fluidigm) was used to view Ct data and amplification curves for the run and export results. The relative abundance of mRNA was normalized to *Ppia*.

**Statistical analysis.** All statistical tests were performed using Prism 8 (GraphPad Software). Points in graphs indicate individual cells (except for speed over time analysis, where points indicate the mean of individual cells), lines indicate the means and error bars indicate the s.e.m. of individual cells. For fluorescence-activated cell sorting analysis, points represent samples, and bar graphs and error bars indicate sample means and s.e.m., respectively. For the intravital imaging experiments, measurements were made on multiple samples (animals) whenever possible (limited motion of the tissue) or necessary (small number of events per sample) and the results were pooled. Otherwise, quantification was performed on a representative sample (animal) and representative results are shown (as indicated in the figure legends). Individual tracks are presented from only one representative experiment for the sake of readability, except for Figs. 1 and 2 and Extended Data Fig. 5 where tracks were pooled from multiple experiments because of the small number of events per sample. The statistical tests employed are detailed in the figure legends. Briefly, for small individual cell numbers or biological samples (<30), a normal distribution was not assumed and nonparametric tests were used systematically. When >30 individual cells were analyzed, the normal distribution was tested and, according to the results, parametric or nonparametric tests were performed.

**Reporting summary.** Further information on research design is available in the Nature Research Reporting Summary linked to this article.

## Data availability
All datasets generated and/or analyzed during the current study are included in this published article. Source data are provided with this paper. The accompanying source data or supplementary information are available from the corresponding author upon reasonable request.

## References
27. Gorelik, R. & Gautreau, A. Quantitative and unbiased analysis of directional persistence in cell migration. *Nat. Protoc.* **9**, 1931–1943 (2014).

## Acknowledgements
We thank G. Eberl, S. K. Durum and O. Mandelboim for the *Rorc*$^{GFP}$, *Il22*$^{TdT}$ and *Ncr1*$^{GFP}$ mice, respectively. At Institut Pasteur, we are grateful to the Central Animal Facility and especially to the animal caretakers, the CB UTechS platform for cytometry support and L. Guenin-Macé (Immunobiology and Therapy Unit) for her help with the permeability assay. We thank all the members of the Innate Immunity and Dynamics of Immune Responses Units for support and helpful discussions, with specific contributions from A. Schiavo, A. Bohineust, R. Thibaut, J. Postat and C. Grandjean. This work is supported by grants from Institut Pasteur, the Institut National de la Santé et de la Recherche Médicale (INSERM), the Agence National pour le Recherche (ANR-15-CE15-0004–ILC3_MEMORY) and the European Research Council (ERC) under the European Union's Horizon 2020 research and innovation program (695467–ILC_REACTIVITY). A.J. is supported by INSERM under an ERC grant and the Fondation pour la Recherche Médicale through the 'Espoirs de la Recherche' program.

## Author contributions
A.J., J.P.D. and N.S. conceived the study. A.J. and N.S. designed and performed the experiments, together with Z.G. for all intravital imaging experiments. S.M. performed the Biomark experiments and analyzed the data. A.D. and I.P. provided the *Ccr9*$^{-/-}$ lymphocytes for the adoptive transfer experiments. A.J. and N.S. analyzed the data and performed the statistical analysis. J.P.D. and P.B. provided key intellectual expertise, resources and acquired the funding. A.J., J.P.D. and N.S. wrote the manuscript. N.S. supervised the study.

## Competing interests
The authors declare no competing interests.

## Additional information
**Extended data** is available for this paper at https://doi.org/10.1038/s41590-022-01284-1.

**Correspondence and requests for materials** should be addressed to Nicolas Serafini.

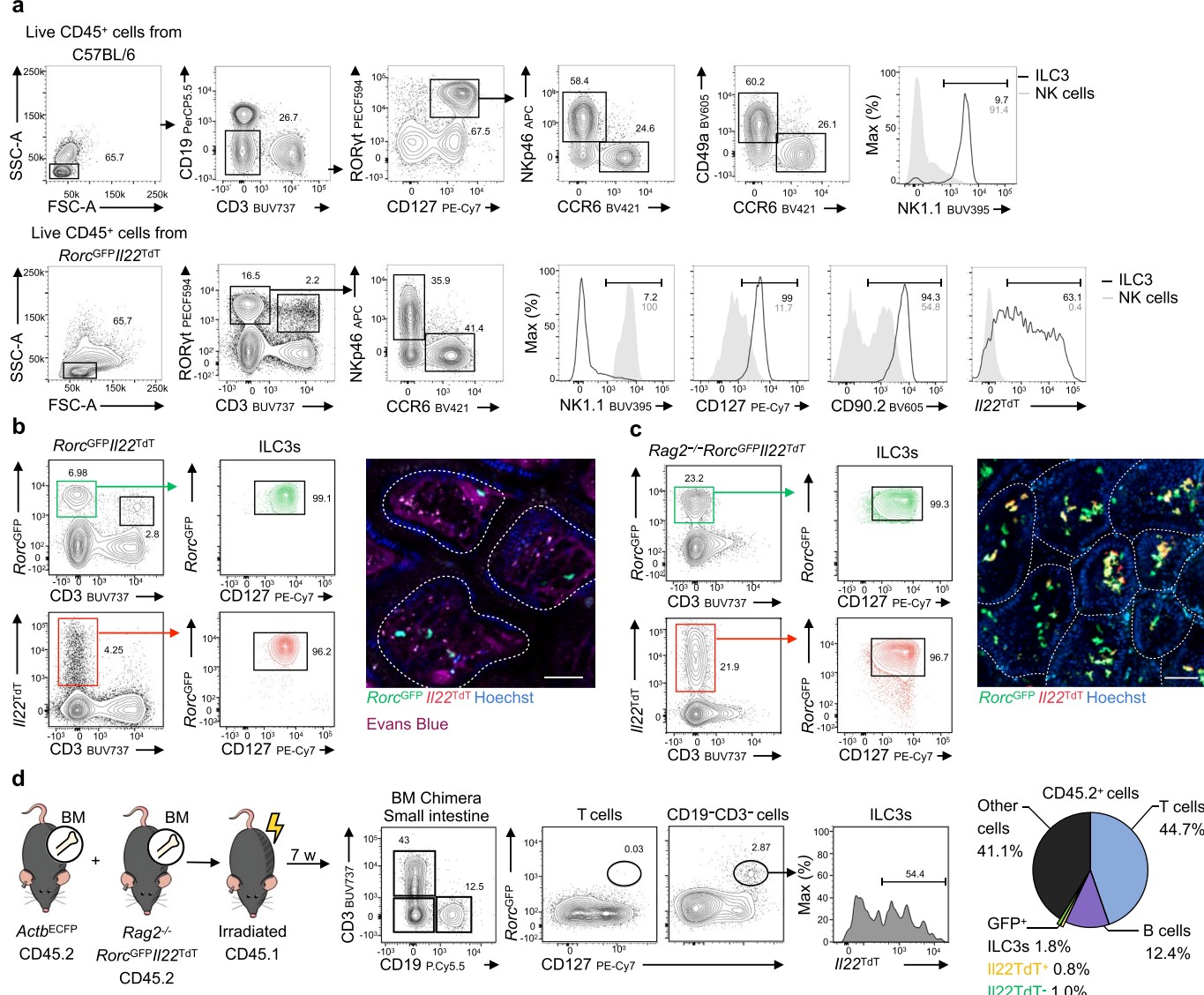

**Extended Data Fig. 1 | BM chimera, a novel model to investigate ILC3 dynamics *in vivo*. a,** Representative gating strategy for identification of ILC3s from small intestine lamina propria of C57BL6 and *Rorc*^GFP^*Il22*^TdT^ mice from one of three independent experiments (ILC3, CD45⁺ CD3⁻ CD19⁻ CD90⁺ CD127⁺ NK1.1⁻ RORγt⁺ cells; NK cells, CD45⁺ CD3⁻ CD19⁻ NKp46⁺ NK1.1⁺ RORγt⁻ cells). **b,** Representative flow cytometry analysis of small intestine from *Rorc*^GFP^*Il22*^TdT^ from one of three independent experiments. Representative intravital image of ILC3s (scale bar, 50 μm) in intestinal villi (V) of *Rorc*^GFP^ *Il22*^TdT^ mice. Mice were injected with Hoechst and Evans blue before imaging to visualize all nuclei (blue) and blood vessels (magenta). Data are representative of three independent experiments. **c,** Representative intravital image (scale bar 50 μm) and flow cytometry analysis of small intestine from *Rag2*^−/−^*Rorc*^GFP^*Il22*^TdT^ reporter mice from one of three independent experiments. **c,** Bone marrow (BM) chimeric mice were created by cell transfer. Mixed bone marrow cells from *Rag2*^−/−^*Rorc*^GFP^*Il22*^TdT^ (95%) and *Actb*^ECFP^ (5%) mice was injected into lethally irradiated congenic C57BL/6 J mice and later analyzed for reconstitution. Representative flow cytometry analysis of reconstitution by donor cells (CD45.2⁺) in the intestinal lamina propria from one of two independent experiments.

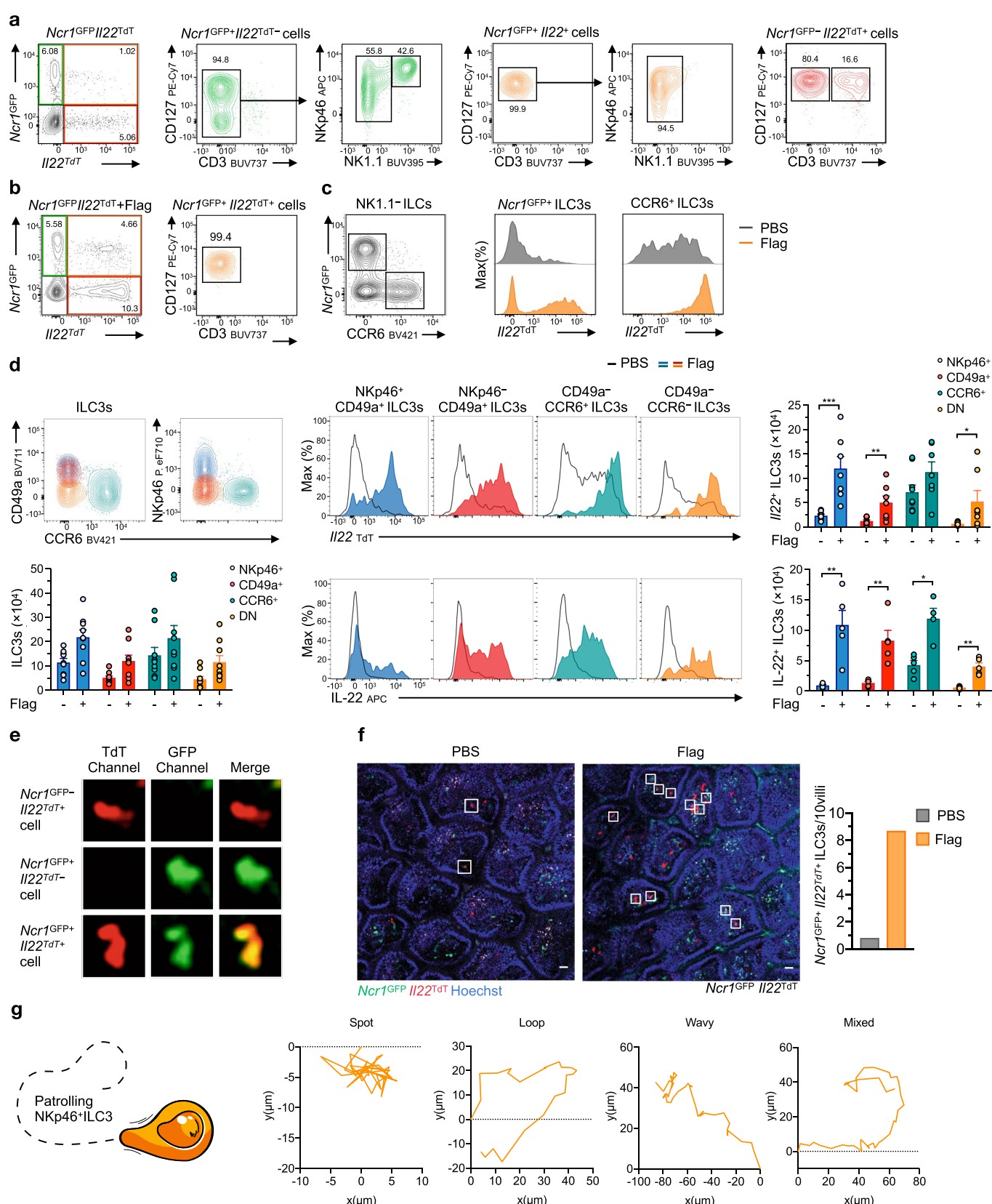

**Extended Data Fig. 2 | See next page for caption.**

**Extended Data Fig. 2 | NKp46[+] ILC3 responses upon bacterial challenge. a**, **b**, Representative flow cytometry analysis of small intestine leukocytes from *Ncr1*[GFP]*Il22*[TdT] mice at indicated conditions from one of three independent experiments. **c**, Representative flow cytometry analysis of *Il22*[TdT] expression in NKp46[+] and CCR6[+] ILC3s (CD45[+] CD3[−] CD5[−] CD90[+] CD127[+] cells) from *Ncr1*[GFP]*Il22*[TdT] mice from one of two independent experiments. **d**, Representative flow cytometry analysis of small intestine ILC3 subsets (pre-gated on CD45[+] CD3[−] CD5[−] NK1.1[−] CD127[+] RORγt[+] cells) from PBS or flagellin (Flag)-treated wild-type mice from one of five experiments. Absolute numbers of total, *Il22*[TdT+] and IL-22[+] ILC3s (n = 9). Each bar corresponds to the mean ± s.e.m of the values obtained (*P < 0.05; **P < 0.01; ***P < 0.001; two-tailed Mann-Whitney test; exact P values are provided in the source data). **e**, Representative images of observable cells in *Ncr1*[GFP]*Il22*[TdT] mice by intravital imaging rom one of three independent experiments. **f**, Representative images and absolute numbers of *Ncr1*[GFP+]*Il22*[TdT+] ILC3s in intestinal villi of from PBS or Flag-treated *Ncr1*[GFP]*Il22*[TdT] mice from one of three independent experiments (scale bar, 30 μm). **g**, Sample tracks of *Ncr1*[GFP+]*Il22*[TdT+] ILC3s patrolling behaviors used in Fig. 2g.

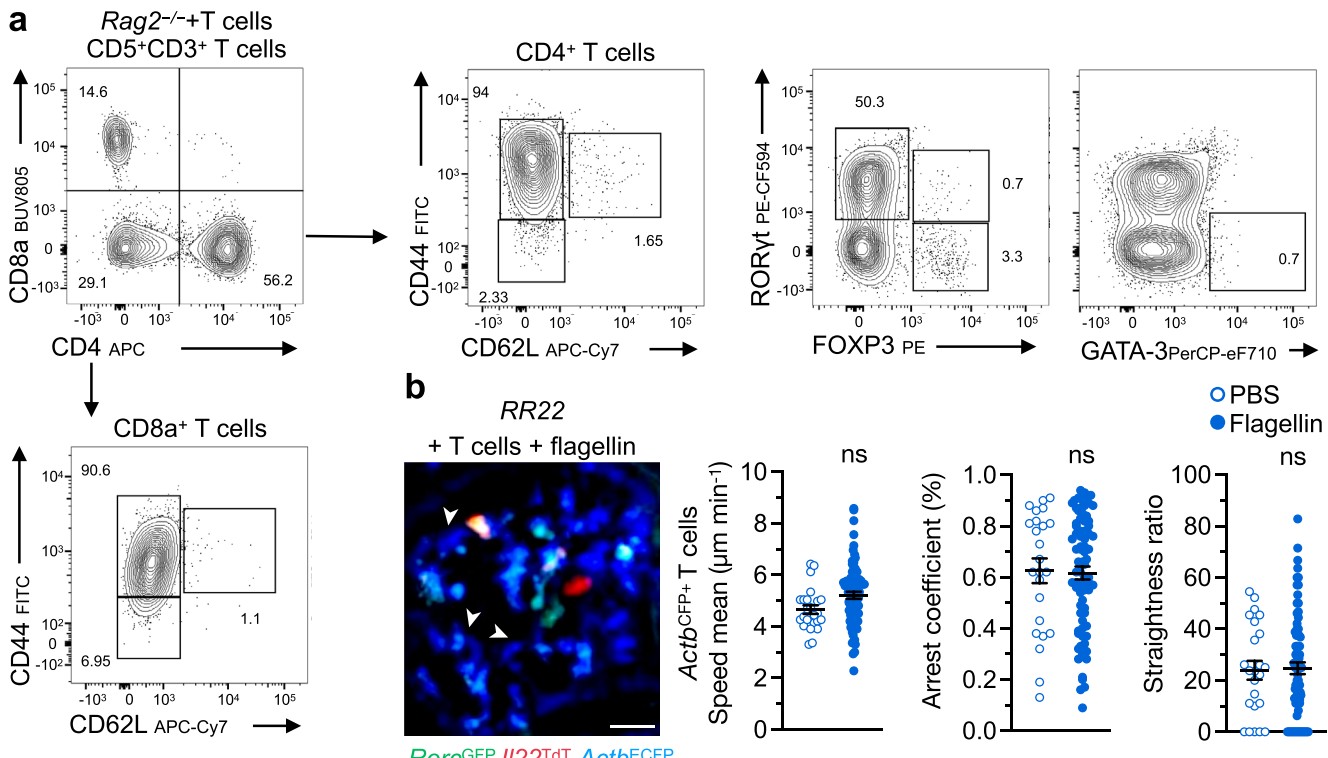

**Extended Data Fig. 3 | Phenotype and dynamics of adoptively transferred intestinal T cells. a**, Representative flow cytometry analysis of T cells in indicated mice. **b**, Representative intravital image (scale bar 20 μm) of $Actb^{ECFP+}$ T cells (white arrowhead) in intestinal villi of indicate mice from one of three independent experiments (scale bar, 40 μm). **c**, Mean speed, arrest coefficient and straightness ratio of adoptively transferred $Actb^{ECFP+}$ T cells in the intestine from treated mice. Results are from at least one movie per condition from one of three independent experiments (n = 24 PBS and 80 Flag). Each bar corresponds to the mean ± s.e.m of the values obtained (ns, not significant; two-tailed Mann-Whitney test; exact P values are provided in the source data). $RR22$, $Rag2^{-/-}Rorc^{GFP}Il22^{TdT}$. Flag, Flagellin.

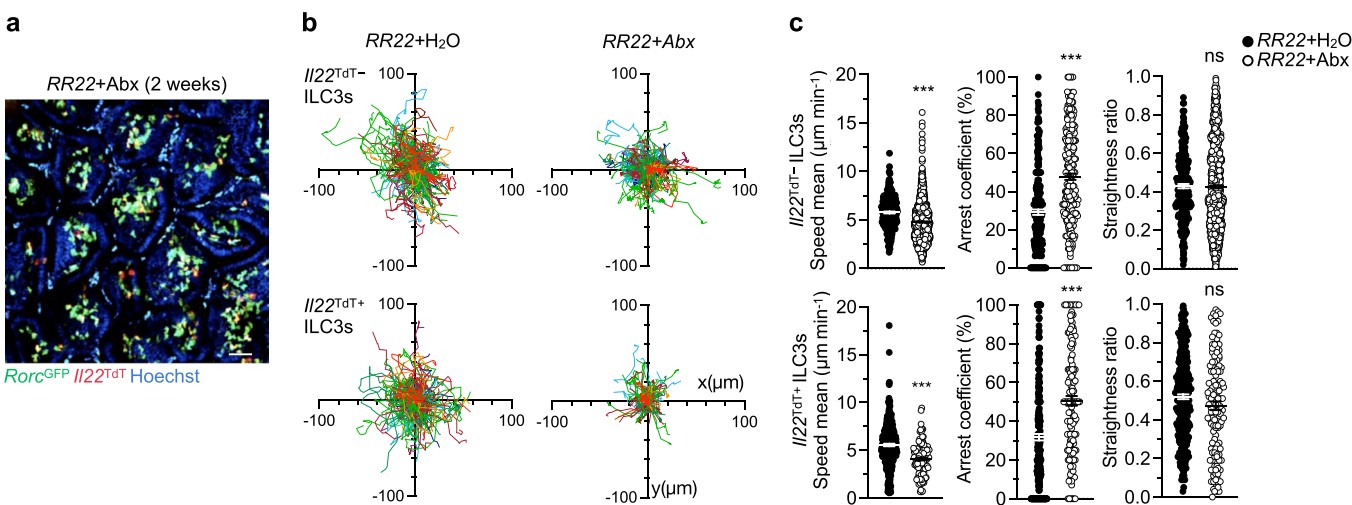

**Extended Data Fig. 4 | Impact of the microbiota on ILC3 patrolling. a**, Representative intravital images of ILC3s (scale bar 50 μm) in intestinal villi Abx-treated *RR22* mice (2 weeks after treatment) from one of three independent experiments. **b**, Individuals tracks of intestinal *Il22*⁻ and *Il22*⁺ ILC3s in treated *RR22* mice. **c**, Graphs show mean speed, arrest coefficient and straightness ratio of indicated populations in *RR22* and *RR22*+Abx mice. Results are from one to three movies per condition from one of three independent experiments (n = 257 and n = 341 for *Il22*⁻ and *Il22*⁺ ILC3s in *RR22* mice, respectively; n = 124 and n = 134 *Il22*⁻ and *Il22*⁺ ILC3s in *RR22* + Abx mice, respectively). Each bar corresponds to the mean ± s.e.m of the values obtained (ns, not significant; ***P < 0.001; one-way ANOVA; exact P values are provided in the source data). *RR22*, *Rag2*⁻/⁻*Rorc*ᴳᶠᴾ*Il22*ᵀᵈᵀ; Abx, antibiotics.

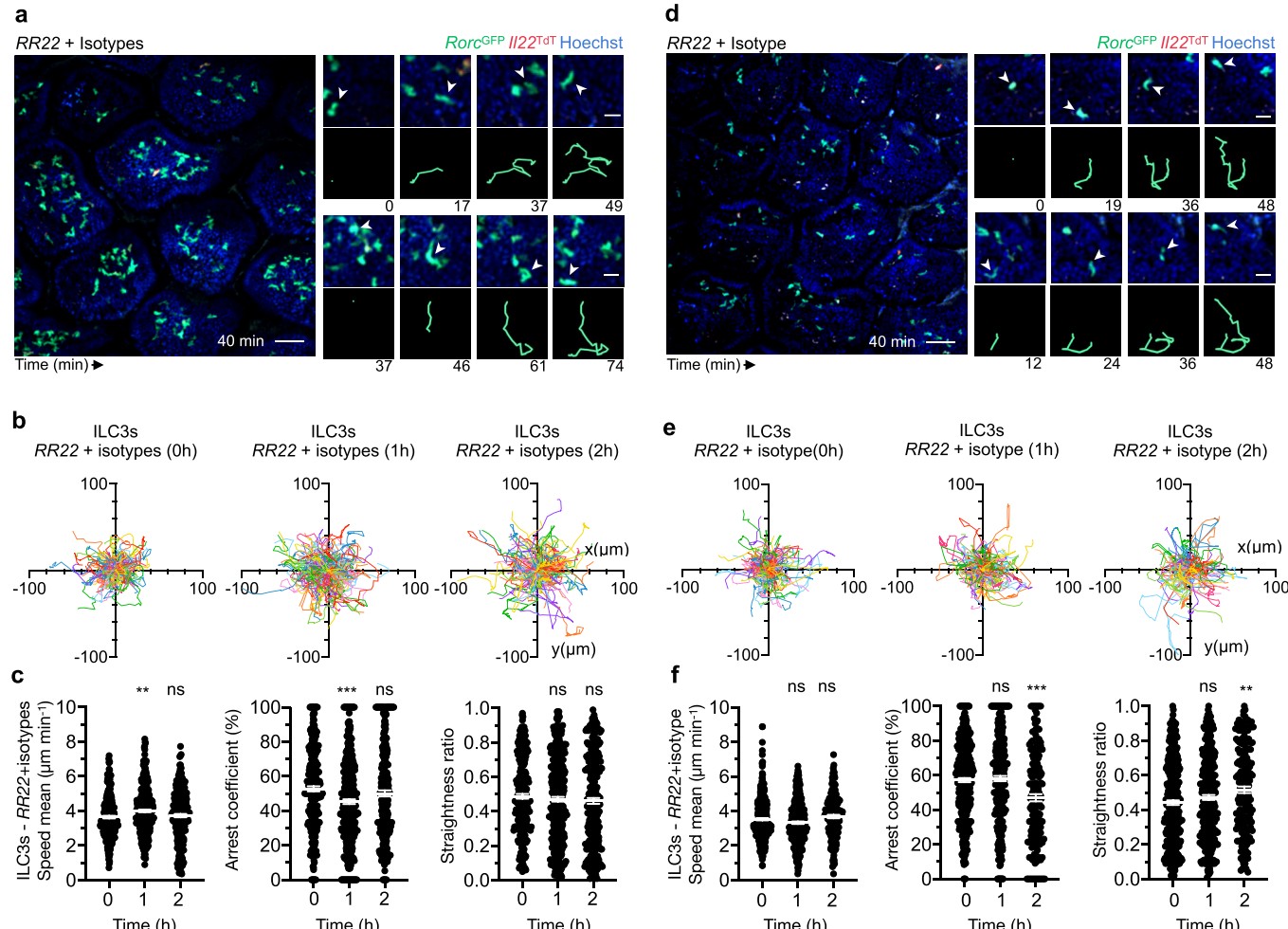

**Extended Data Fig. 5 | Isotypes control have little effect on ILC3 migration. a-f,** *RR22* mice were injected *i.v.* with combination of isotypes in (**a-c**): mouse IgG1, rat IgG2a and IgG2b, or in (**d**-**f**) rat IgG2b only. Representative image (left; scale bar 50 μm) and time-lapse images (right; scale bar 15 μm) of ILC3 in intestinal villi after isotype controls injection at 0 min (combination in **c** or rat IgG2b alone in **f**). Data are representative of three independent experiments. Individual tracks of intestinal ILC3s before (-60-0 min), 1h (0–60 min) and 2 h (60–120 min) after isotype controls injection (combination in **d** or rat IgG2b alone in **g**) of *RR22* mice. Mean speed, arrest coefficient and straightness ratio of intestinal ILC3s at indicated time points after isotype controls (combination in **e** or rat IgG2b alone in **h**) injection. Results in (**d-e**) are from two to four movies per condition obtained in two independent experiments (n = 266, 0 h; n = 340, 1h; n = 259, 2 h). Results in (**g-h**) are from three to four movies per condition obtained in two independent experiments (n = 349, 0 h; n = 307, 1h; n = 178, 2 h). Each bar corresponds to the mean ± s.e.m of the values obtained (ns, not significant; \*\*$P < 0.002$, \*\*\*$P < 0.0001$; one-way ANOVA; exact P values are provided in the source data).

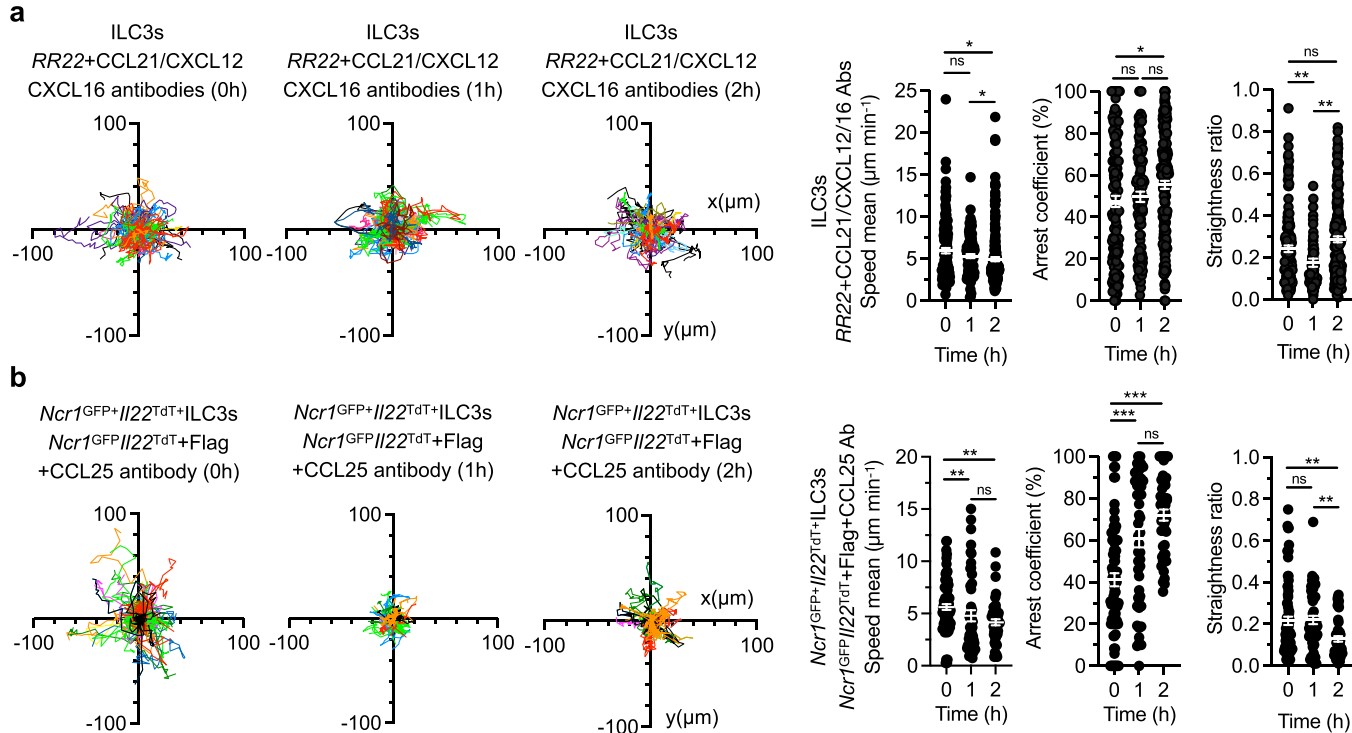

**Extended Data Fig. 6 | Identification of CCL25 as an essential chemokine controlling ILC3 motility. a**, Intestinal ILC3s were imaged in *RR22* mice using intravital imaging for one hour. Then, mice were injected *i.v.* with a combination of blocking monoclonal antibodies (anti-CXCL12, anti-CXCL16, and anti-CCL21) and subsequently imaged for 2 hours. Individual tracks of intestinal ILC3s before (−60–0 min), 1h (0–60 min) and 2 h (60–120 min) after blocking antibodies injection. Mean speed, arrest coefficient and straightness ratio of intestinal ILC3s at indicated timepoints after blocking antibodies injection. Results in (**a**) are from at least five movies per condition obtained in two independent experiments (n = 129, n = 100, and n = 184 ILC3s for 0 h, 1h and 2 h, respectively). **b**, Individual tracks of intestinal NKp46+*Il22*+ ILC3s before (−60–0 min), 1h (0–60 min) and 2 h (60–120 min) after CCL25 Ab injection and 5 h after Flag injection. Mean speed, arrest coefficient and straightness ratio of intestinal NKp46+ *Il22*+ ILC3s at indicated timepoints after CCL25 Ab injection. Results in (**b**) are from at least five movies per condition obtained in two independent experiments (n = 78, 0 h; n = 50, 1h; n = 49, 2 h). Each bar corresponds to the mean ± s.e.m of the values obtained (ns, not significant; *P < 0.05, **P < 0.01, ***P < 0.001; one-way ANOVA; exact P values are provided in the source data). *RR22, Rag2*−/−*Rorc*GFP*Il22*TdT; CCL25 Ab, CCL25 blocking antibody; Flag, Flagellin.

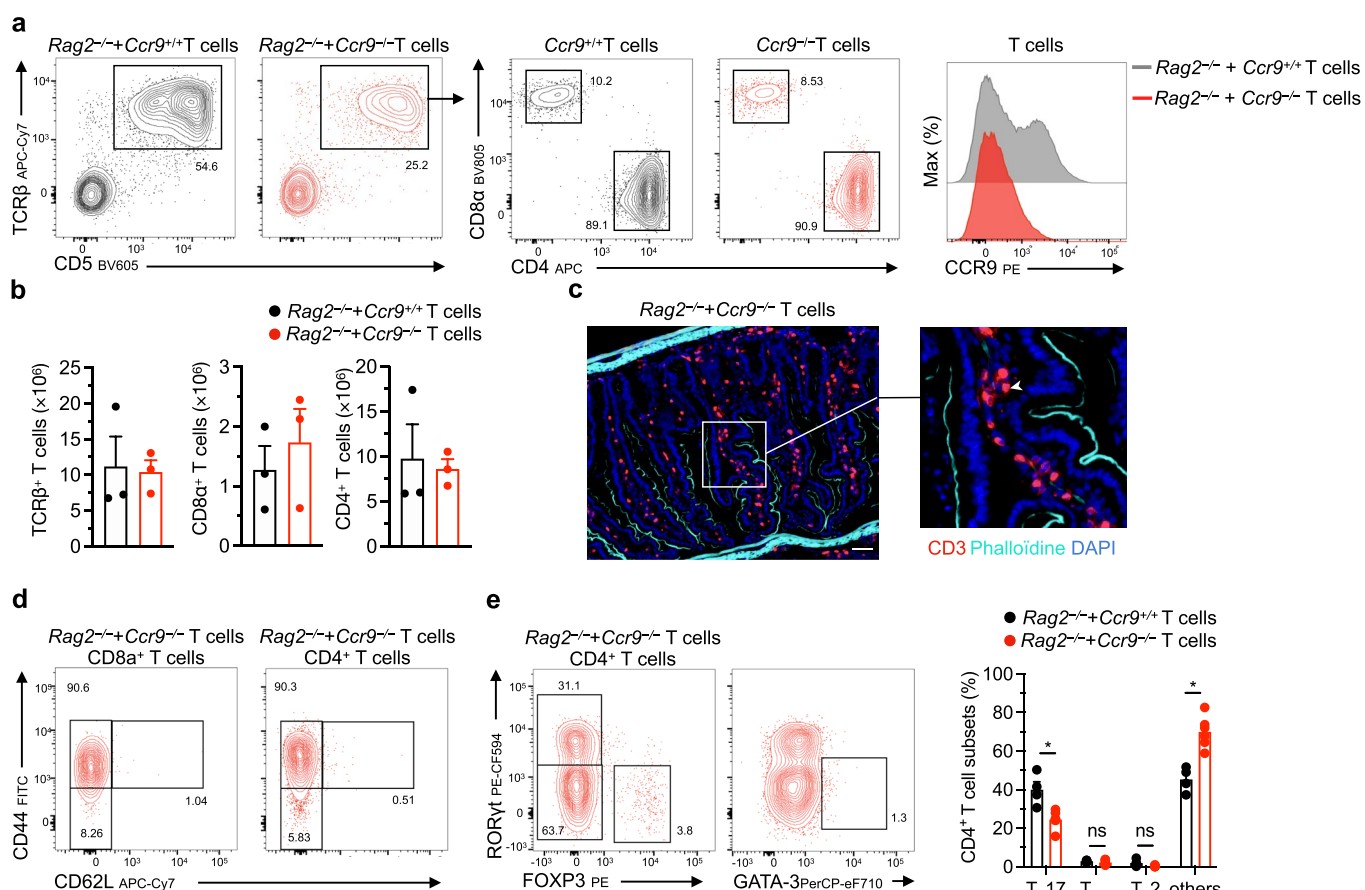

**Extended Data Fig. 7 | *Ccr9*$^{-/-}$ T cells home into the intestinal tissue after adoptive transfer. a**, Representative flow cytometry analysis of intestinal T cells in *Rag2*$^{-/-}$ mice two weeks post-adoptive transfer of either *Ccr9*$^{+/+}$ or *Ccr9*$^{-/-}$ T cells from one of two independent experiments. **b**, Absolute numbers of total TCRβ$^+$ T cells, as well as CD8α$^+$ and CD4$^+$ T cells subsets, in the small intestine of *Rag2*$^{-/-}$ mice two weeks post-adoptive transfer of either control *Ccr9*$^{+/+}$ (n = 3) or *Ccr9*$^{-/-}$ (n = 3) T cells. **c**, Immunofluorescence of ileum from *Rag2*$^{-/-}$ mice adoptively transferred with *Ccr9*$^{-/-}$ T cells (scale bar 50 μm). Data are representative of two independent experiments. **d-e**, Representative flow cytometry analysis of intestinal CD8a$^+$ (**d**) and CD4$^+$ (**e**) T cell subsets in *Rag2*$^{-/-}$ mice two weeks post adoptive transfer of *Ccr9*$^{-/-}$ T cells from one of two independent experiments. Frequencies of CD4$^+$ T cells in the small intestine of *Rag2*$^{-/-}$ mice two weeks post-adoptive transfer of either control *Ccr9*$^{+/+}$ (n = 4) or *Ccr9*$^{-/-}$ (n = 6) T cells. Results in were obtained in three independent experiments. Each bar corresponds to the mean ± s.e.m of the values obtained (ns, not significant; *$P$ < 0.05; two-tailed Mann-Whitney test; exact P values are provided in the source data).

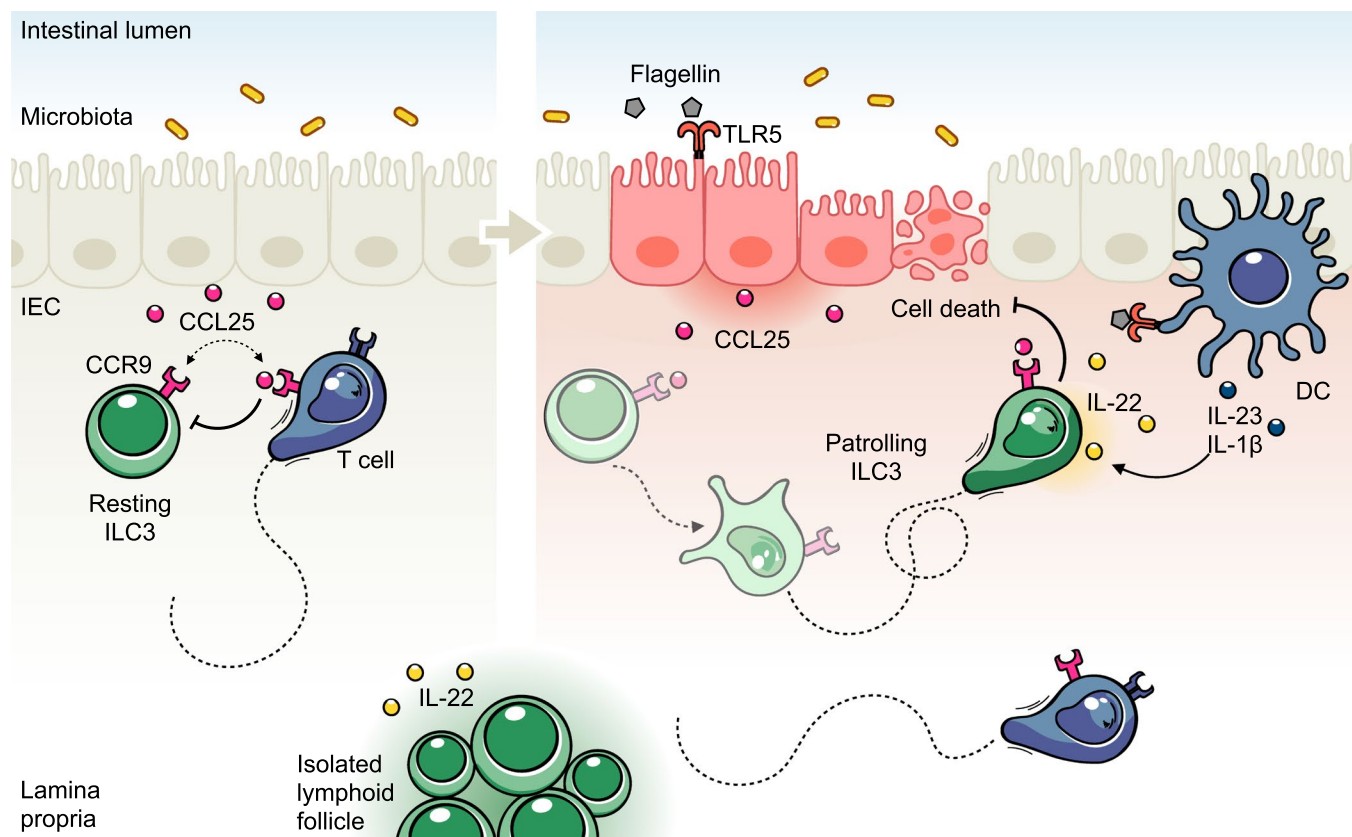

**Extended Data Fig. 8 | Monitoring of intestinal tissue by patrolling ILC3s.** The intestinal mucosa is continuously exposed to environmental stimuli that can induce immune responses through adaptation of local immune cells, including ILC3s. Under basal conditions, villus T cells control ILC3 patrolling though competition for the chemokine CCL25, and consequently villus ILC3s are largely immotile. Disruption of intestinal homeostasis, notably during TLR5-mediated inflammation, leads to the activation of ILC3s and to CCL25-mediated ILC3 patrolling of the intestinal barrier. Intestinal ILC3s produce IL-22 which is critical to prevent IEC death and to maintain the integrity of the intestinal barrier. Tissue scanning of patrolling ILC3s within intestinal villi could be important to support elevated IL-22 concentrations close to IECs and for optimal induction of their functional programs in order to maintain intestinal integrity. Collectively, tissue environmental signals shape intestinal ILC3 activity, including patrolling behavior and IL-22 production, to promote appropriate immune responses and intestinal barrier function. ILC3, group 3 innate lymphoid cell; DC, dendritic cell. CCL25, chemokine ligand 25; TLR-5, Toll-like receptor 5; IL-22, interleukin 22; IEC, intestinal epithelial cell.

# Reporting Summary

## Statistics

For all statistical analyses, confirm that the following items are present in the figure legend, table legend, main text, or Methods section.

| n/a | Confirmed | |
|---|---|---|
| ☐ | ☒ | The exact sample size (*n*) for each experimental group/condition, given as a discrete number and unit of measurement |
| ☐ | ☒ | A statement on whether measurements were taken from distinct samples or whether the same sample was measured repeatedly |
| ☐ | ☒ | The statistical test(s) used AND whether they are one- or two-sided *Only common tests should be described solely by name; describe more complex techniques in the Methods section.* |
| ☒ | ☐ | A description of all covariates tested |
| ☐ | ☒ | A description of any assumptions or corrections, such as tests of normality and adjustment for multiple comparisons |
| ☐ | ☒ | A full description of the statistical parameters including central tendency (e.g. means) or other basic estimates (e.g. regression coefficient) AND variation (e.g. standard deviation) or associated estimates of uncertainty (e.g. confidence intervals) |
| ☐ | ☒ | For null hypothesis testing, the test statistic (e.g. *F*, *t*, *r*) with confidence intervals, effect sizes, degrees of freedom and *P* value noted *Give P values as exact values whenever suitable.* |
| ☒ | ☐ | For Bayesian analysis, information on the choice of priors and Markov chain Monte Carlo settings |
| ☒ | ☐ | For hierarchical and complex designs, identification of the appropriate level for tests and full reporting of outcomes |
| ☒ | ☐ | Estimates of effect sizes (e.g. Cohen's *d*, Pearson's *r*), indicating how they were calculated |

*Our web collection on statistics for biologists contains articles on many of the points above.*

## Software and code

Policy information about availability of computer code

| Data collection | Intravital imaging: Olympus FV31S-SW 2 (Olympus)<br>Flow cytometry: BD FACSDiva 7 and 8 (BD Biosciences)<br>Biomark: Biomark HD Data Collection 4 (Fluidigm)<br>Immunofluorescence: Imager Z.2 (Zeiss)<br>Fluorescence: Fluostar Optima (BMG labtech) |
|---|---|
| Data analysis | Intravital imaging: Imaris 7.4.2 (Bitplane), Fiji 2 (ImageJ), DiPer (PMID: 25033209)<br>Flow cytometry: FlowJo 10 (TreeStar)<br>Biomark: Fluidigm Real-Time PCR Analysis 4.5.2 (Fluidigm)<br>Fluorescence: BMG labtech software (v2)<br>All: Prism 8 (GraphPad) and Excel 16 (Microsoft) |

For manuscripts utilizing custom algorithms or software that are central to the research but not yet described in published literature, software must be made available to editors and reviewers. We strongly encourage code deposition in a community repository (e.g. GitHub). See the Nature Portfolio guidelines for submitting code & software for further information.

## Data

Policy information about availability of data

All manuscripts must include a data availability statement. This statement should provide the following information, where applicable:
- Accession codes, unique identifiers, or web links for publicly available datasets
- A description of any restrictions on data availability
- For clinical datasets or third party data, please ensure that the statement adheres to our policy

> All data sets generated and/or analyzed during the current study are included in this published article. The accompanying source data or supplementary information are available from the corresponding author upon reasonable request. Source data are provided with this paper.

## Human research participants

Policy information about studies involving human research participants and Sex and Gender in Research.

| | |
|---|---|
| Reporting on sex and gender | n/a |
| Population characteristics | n/a |
| Recruitment | n/a |
| Ethics oversight | n/a |

Note that full information on the approval of the study protocol must also be provided in the manuscript.

# Field-specific reporting

Please select the one below that is the best fit for your research. If you are not sure, read the appropriate sections before making your selection.

☒ Life sciences    ☐ Behavioural & social sciences    ☐ Ecological, evolutionary & environmental sciences

For a reference copy of the document with all sections, see nature.com/documents/nr-reporting-summary-flat.pdf

# Life sciences study design

All studies must disclose on these points even when the disclosure is negative.

| | |
|---|---|
| Sample size | Sample sizes were derived from initial preliminary experiments and from a principle of repeating all in vivo experiments at least twice to ensure experimental reproducibilty. No statistical methods were used to pre-determine samples sizes. For intravital imaging experiments, external parameters were controlled (temperature, oxygen), experimental bias was limited (no sample preparation besides surgery), multiple XY positions could be simultaneously acquired and the effect size was large – therefore a small sample size (n=3 in general) was powerful enough. For flow cytometry and Biomark experiments that involved sample preparation, sample size was of n=4 minimum (and in general > n=6) and this sample size was powerful enough too. |
| Data exclusions | No data were excluded from the analyses. |
| Replication | All the experiments were reliably reproduced as validated by at least two independent experiments. Moreover, two distinct investigators performed the experiments independently. For each technique, two distinct investigators also performed independent data analysis on an identical dataset. All attempts of replication were successful. |
| Randomization | Experimental groups were not randomized. Mice were grouped according to genotype and all experiments were performed with age- and sex-matched littermates. The sex-specific histocompatibility was observed for adoptive transfer. |
| Blinding | Investigators were not blinded during the acquisition of the data and analysis because 1) the investigator analyzing data was also collecting it 2) experimental groups could be determined from the data (e.g. images from intravital imaging where adoptive transfer could be deducted from CFP+ T cells that could be visualized on images). |

# Reporting for specific materials, systems and methods

We require information from authors about some types of materials, experimental systems and methods used in many studies. Here, indicate whether each material, system or method listed is relevant to your study. If you are not sure if a list item applies to your research, read the appropriate section before selecting a response.

## Materials & experimental systems

| n/a | Involved in the study |
|---|---|
| ☐ | ☒ Antibodies |
| ☒ | ☐ Eukaryotic cell lines |
| ☒ | ☐ Palaeontology and archaeology |
| ☐ | ☒ Animals and other organisms |
| ☒ | ☐ Clinical data |
| ☒ | ☐ Dual use research of concern |

## Methods

| n/a | Involved in the study |
|---|---|
| ☒ | ☐ ChIP-seq |
| ☐ | ☒ Flow cytometry |
| ☒ | ☐ MRI-based neuroimaging |

## Antibodies

Antibodies used: The following antibodies were used for the study (all commercially available): CCR2 BV421 (SA203G11, Biolegend), CCR6 BV421 (140706, BD Biosciences), CCR7 PE (4B12, eBioscience), CCR9 PE (CW-1.2, eBioscience), CD11b AF700 (M1/70, eBioscience), CD19 PerCP-Cy5.5 (1D3, BD Biosciences), CD127 PE-Cy7 (A7R34, eBioscience), CD3 APC (145-2C11, eBioscience), CD3 eF450 (500A2, eBioscience), CD3 BUV737 (17A2, BD Biosciences), CD44 (IM7, BD Biosciences), CD62L (MEL-14, BD Biosciences), Foxp3 (FJK-16s, eBioscience), Gata3 (TWAJ, eBioscience), active Casp3 (C92-605, BD Biosciences), CD45 BUV395 (30-F11, BD Biosciences), CD45.1 APC-Cy7 (A20, BD Biosciences), CD45.2 APC- eF780 (104, eBioscience), CD45.2 BV711 (104, BD Biosciences), CD45.2 FITC (104, BD Biosciences), CD5 APC-R700 (53-7.3, BD Biosciences), CD5 BV605 (53-7.3, BD Biosciences), CD8α BUV805 (53-6.7, BD Biosciences), CD4 APC (GK1.5, eBioscience), CD90.2 PerCP-eF710 (30-H12, eBioscience), CD90.2 BV605 (53-2.1, Biolegend), CD90.2 FITC (53-2.1, eBioscience), CXCR3 BV421 (CXCR3-173, BD Biosciences), CXCR4 PE (QA16A08, Biolegend), CXCR5 PE (L138D7, Biolegend), CXCR6 PE (SA051D1, Biolegend), KLRG1 BV605 (2F1, BD Biosciences), NK1.1 APC-Cy7 (PK136, Biolegend), NK1.1 BUV395 (PK136, BD Biosciences), NKp46 APC (29A1.4, eBioscience), NKp46 PE (29A1.4, eBioscience), RORgt PE-CF594 (Q31-378, BD Biosciences), TCRβ APC-eF780 (H57-597, eBioscience).
All antibodies were used at 1:200, except for CD90.2 was used at 1:800, and CCR2/CXCR3/4/5/6 were used at 1:100.

Validation: All antibodies are commercially available and are validated on the manufacturer's website. All antibodies were initially tested and titrated before routine use in the lab.

## Animals and other research organisms

Policy information about studies involving animals; ARRIVE guidelines recommended for reporting animal research, and Sex and Gender in Research

Laboratory animals: Mice were housed and bred in dedicated in animal facilities of the Institut Pasteur (12h light/dark cycle; 22+/-2°C) . All mice were used on a C57BL/6J background. Eight to fourteen weeks-old male and female animals, except for experiments on BM chimeric mice which nvolved sixteen weeks-old males at the time of analysis.

Wild animals: The study did not involve wild animals.

Reporting on sex: All experiments were performed on eight to fourteen weeks-old male and female animals, except for experiments on BM chimeric mice which involved sixteen weeks-old males at the time of analysis.

Field-collected samples: The study did not involve samples collected from the field.

Ethics oversight: All experiments involving mice were performed according to guidelines issued by the Institut Pasteur Ethics Committee and were approved by the French Ministry of Research (projects dha170001, CETEA 2013-033 and CETEA 17500).

Note that full information on the approval of the study protocol must also be provided in the manuscript.

## Flow Cytometry

### Plots

Confirm that:

☒ The axis labels state the marker and fluorochrome used (e.g. CD4-FITC).

☒ The axis scales are clearly visible. Include numbers along axes only for bottom left plot of group (a 'group' is an analysis of identical markers).

☒ All plots are contour plots with outliers or pseudocolor plots.

☒ A numerical value for number of cells or percentage (with statistics) is provided.

### Methodology

Sample preparation: Small intestine was collected from euthanized mice and placed into cold complete medium: RPMI 1640 GlutaMAX (Gibco) supplemented with 5% fetal calf serum (Eurobio) and 10mM HEPES (Sigma-Aldrich). The mesenteric adipose tissue and Peyer's patches were first pulled out before cutting the small intestine longitudinally and removing feces. Intestinal tissue

was washed in PBS (Gibco) to eliminate mucus, cut into 1-2 cm pieces and intestinal epithelial cells were eliminated by shaking incubation in complete medium containing 5 mM EDTA (Invitrogen) for 20 min at 37°C. Subsequently, the intestinal tissue was minced and incubated twice in a digestion solution containing 0.5 mg Collagenase VII (Sigma-Aldrich) for 15 min at 37°C in a shaking incubator to isolate the lamina propria lymphocytes (LPL). LPL were filtered through a 40-μm cell strainer and kept in complete medium for downstream analysis.

| | |
|---|---|
| Instrument | All the samples were acquired on a custom-configured LSR Fortessa (BD Biosciences) or sorted using a custom-configured FACSAria III (BD Biosciences). |
| Software | Data were collected with BD FACSDiva 7 and 8 (LSR Fortessa, BD Biosciences) and analyzed with FlowJo10 (TreeStar). |
| Cell population abundance | The purities of sorted ILC3 were more than 98 %, as determined by flow cytometry analysis on the sample containing purified ILC3. |
| Gating strategy | For all experiments, FSC-A versus SSC-A gates of the starting cell population were used to identify cells. Next, FSC-H versus FSC-W and SSC-H versus SSC-W gates were used to identify singlets cells. Flexible Viability Dye staining was then used to identify live cells. Positive populations were determined by the specific antibodies staining, which were distinct from negative populations. Specifically, for the analysis of intestinal lymphocytes, we applied the following gating strategies: FSC-A versus SSC-A / FSC-H versus FSC-W / SSC-H versus SSC-W / Live (Flexible Viability Dye−) and then<br>- CD45.2+ / CD3− CD19− / RorcGFP+ CD127+ for ILC3, CD45.2+ / CD3+ for T cells and CD45.2+ / CD19+ for B cells in BM chimeric mice<br>- CD45+ / Ncr1GFP versus Il22TdT and notably CD45 / Ncr1GFP+ Il22TdT+ / CD127+ CD3− / NKp46+ NK1.1−/lo for NKp46+ Il22+ ILC3 in Ncr1GFP Il22TdT mice<br>- CD45.2+ / RorcGFP+ CD3− / RorcGFP+ CD127+ for ILC3, CD45.2+ / RorcGFP+ CD3+ for TH17/22, CD45.2+ / Il22TdT+ CD3− / RorcGFP+ CD127+ for Il22+ ILC3 in (Rag2−/−) RorcGFP Il22TdT mice<br>- CD45+ / CD3− CD5− / CD90+CD127+ / NK1.1− for ILC NK1.1−, sub-gated on Ncr1GFP and CCR6 for NKp46+ or CCR6+ ILC3 respectively in Ncr1GFP Il22TdT mice<br>- CD45.2+ / CD3− CD5− / CD19− / CD90+CD127+ / RORγt+ and sub-gated on NKp46 and CCR6 for NKp46+ or CCR6+ ILC3 respectively, CD45.2+ / CD3+ CD5+ for T cells, and for control populations CD45.2+ / CD3+ NK1.1+ for NKT cells, CD45.2+ / CD19+ for B cells, CD45.2+ / CD3− CD5− / CD19− / CD11c+ for myeloid cells, in C57BL6/J mice (chemokine receptors analysis)<br>- CD45+ CD11b−/ TCRβ+ CD5+ for TCRβ+ T cells and sub-gated on CD8α and CD4 for CD8α+ and CD4+ T cells respectively<br>For the sorting of ILC3 subsets, we applied the following gating strategies: FSC-A versus SSC-A / FSC-H versus FSC-W / SSC-H versus SSC-W / Live (Flexible Viability Dye−) / CD45.2+ / CD3− / NK1.1− / KLRG1− / CD90.2+CD127+ and sub-gated on NKp46+ or CCR6+ cells. |

☒ Tick this box to confirm that a figure exemplifying the gating strategy is provided in the Supplementary Information.

