## [Peer Review File · Nature Immunology]

Peer Review Information

Journal: Nature Immunology

Manuscript Title: Inflammation triggers ILC3 patrolling of the intestinal barrier

Corresponding author name(s): Nicolas Serafini

Editorial Notes:

Reviewer Comments & Decisions:

Decision Letter, initial version:
--

7th Oct 2021

Dear Dr. Serafini,

Your Letter, "Inflammation triggers ILC3 patrolling of the intestinal barrier" has now been seen by 2 referees. While we find your work of considerable potential interest, the reviewers have raised substantial concerns that must be addressed. As such, we cannot accept the current version of the manuscript for publication, but would be happy to consider a revised version that addresses these concerns, as long as novelty is not compromised in the interim.

Please revise to address the issues raised by the referees. We do not consider it is essential to provide data on the motility of ILCs in other tissues. At resubmission, please include a point-by-point "Response to referees" detailing how you have addressed each referee comment (please specify page and figure number). This response will be sent back to the referees along with the revised manuscript.

In addition, please include a revised version of any required reporting checklist. It will be available to referees (and, potentially, statisticians) to aid in their evaluation if the manuscript goes back for peer review. A revised checklist is essential for re-review of the paper.

The Reporting Summary can be found here:

we hope to receive the revised manuscript within 6 months. If you cannot send it within this time, please let us know. We will be happy to consider your revision so long as nothing similar has been accepted for publication at Nature Immunology or published elsewhere.

Nature Immunology is committed to improving transparency in authorship. As part of our efforts in this direction, we are now requesting that all authors identified as 'corresponding author' on published papers create and link their Open Researcher and Contributor Identifier (ORCID) with their account on the Manuscript Tracking System (MTS), prior to acceptance. ORCID helps the scientific community achieve unambiguous attribution of all scholarly contributions. You can create and link your ORCID from the home page of the MTS by clicking on 'Modify my Springer Nature account'. For more information please visit www.springernature.com/orcid.

Thank you for the opportunity to review your work.

Sincerely,

Ioana Visan, Ph.D.
Senior Editor
Nature Immunology

Tel: 212-726-9207
Fax: 212-696-9752
www.nature.com/ni

Reviewers' Comments:

Reviewer #1:

Remarks to the Author:

The manuscript by Jarade et al., entitled "Inflammation triggers ILC3 patrolling of the intestinal barrier" seeks to investigate the behaviour of ILC3 populations within the small intestine using refined in vivo imaging approaches. After 10+ years of experiments revealing the requirement for ILC3 in intestinal homeostasis and defence, their actual behaviour within this tissue, which must contribute to how their roles are fulfilled, remains very poorly understood. Thus, I would concur with the authors that this is a key knowledge gap and to my knowledge this is the first study of its kind to really tackle this convincingly and enhance understanding of ILC migratory behaviour within the intestine. I think the findings are of clear interest to both mucosal immunologists but also more broadly across the immunology spectrum given the intense focus on ILC functions following their relatively recent discovery.

The manuscript is concise and clear in its basic discoveries, describing differential ILC3 behaviour associated with the two major ILC3 subsets present within the SILP (NCR+ and LTI-like/CCR6+). While the location of these populations was indicated by previous studies, a key initial discovery of this manuscript is that ILC3 populations are surprisingly static at 'steady state'. The authors further show that expression of IL-22 is largely restricted to the LTI-like population located within cryptopatches. To assess the migratory behaviour of ILC3, the authors employ the clever use of novel mice and bone marrow chimeras. This convincingly circumvents challenges in specific identification of ILC in the absence of T cell 'contamination' and overcomes the technical issues that I can envisage in definitively assessing ILC3 in vivo. The authors further provide evidence that acute inflammation drives altered behaviour within the villi ILC3 population inducing a patrolling phenotype which is clearly described. However, the

relevance of i.v. flagellin versus local gut stimuli is a weak point in the study. The authors go on to show that the presence of an adaptive immune system impacts migration and provide reasonable evidence that availability of CCL25 controls the migratory ILC3 behaviour. Collectively these data provide a detailed and novel characterisation of the local migratory characteristics of ILC3 within the small intestine. In addition to some controls (expanded on below) to aid understanding of the experiments performed, where the manuscript is currently weaker is defining what the patrolling phenotype really means for the local intestinal environment and the extent to which the static ILC behaviour is tissue specific or a basic feature of these cells regardless of where they reside. To develop the manuscript beyond a novel and interesting description, evidence of how inducing a patrolling behaviour benefits intestinal defence would build more functional relevance into the study. Given the extensive data linking ILC3 to regulation of intestinal T cell responses, the behaviour of the ILC3 within draining lymphoid tissue exploiting the clever imaging approaches used here would further advance the field.

Major comments

1. i.v. flagellin is used to induce a patrolling behaviour. While the data is clear, this systemic intervention should be contrasted with stimulations that better target the intestinal tract and provide more physiological insight into ILC3 behaviour and how this is altered. While not perfectly aligned to the small intestine, infection models such as Citrobacter or Salmonella would seem potentially appropriate to further investigate the behavioural responses of ILC in the small intestine in response to more local stimulation. Other options might be orally administered R848 + Ag.
2. An obvious question raised by the altered behaviour of the ILC3 within the villi is what this means for the local environment. What changes within the villi when ILC3 patrol rather than remain static? What is the benefit here? Given that at steady state the authors provide evidence that IL-22 expression is restricted to the cryptopatch ILC3s, this suggests that IL-22 readily diffuses to support epithelial integrity in the gut. Is there data that IL-22 only acts very locally and thus patrolling helps disperse IL-22 at key locations? What then is the advantage of patrolling if IL-22 can diffuse? Perhaps other ILC3 functions are more relevant and IL-22 expression is a read-out but not the key function?
3. Can the authors provide evidence that the altered behaviour in (i.e. patrolling in steady state) in Rag^{-/-} mice is not due to an altered microbiome and different stimuli feeding back into the system (i.e. rather than the absence of T cell down modulation of patrolling). It seems highly likely the microbiome is substantially altered in the absence of T cells. This potential confounder is not touched upon and some data to support that this is not microbiome driven would enhance the data.
4. The impact of the observations here would be enhanced by determining if ILC3 in other tissue locations are static. Appreciating that not all tissue compartments lend themselves to imaging, the mLN and ILC3 behaviour within the interfollicular zone would be an obvious place to look, given the specific location in this region (e.g. Mackley et al., 2015) and the link to MHCII-dependent regulation of T cell

responses (e.g. Hepworth/Sonnenberg studies). TO my knowledge, there is currently no data on the behaviour of ILC3 within the interfollicular region where they may migrate with a defined region or be more static and let activated T cells come to them. The authors have the tools to address this and the basic question of where ILC3 regulate responses (i.e. LN vs gut LP) remains unclear.

Minor comments:

1. In Fig. 4b – is the CCR9 expression show in WT or Rag^{-/-} mice. Is CCR(more abundantly expressed in Rag^{-/-} ILC3? There is only some limited CCR9 expression shown on the NKp46 compartment. Does this indicate that only some ILC3 can patrol? Do CCR9^{-/-} ILC3 still patrol?
2. In Fig. 3b, how well is the intestine reconstituted with T cells after transfer into the 'RR22' mice? What is the phenotype of the T cells here? There is some evidence from a limited number of mice (Ext Figure 5) that total T cell numbers are not significantly different. The data is somewhat limited. Is the phenotype of the T cells the same post-reconstitution? Given the role of CCR9 in SILP homing, is surprising that there is no impact.
3. In Fig. 2c is the total ILC compartment increase or are simply more NCR⁺ ILC3 turning on IL-22? This should be clarified – would be easy to show total numbers of the ILC3 subsets for example.
4. How does flagellin administration impact the cryptopatch ILC compartment? This isn't commented on even though the expression of IL-22 by the cryptopatch ILC3 (CCR6⁺) appears upregulated (Ext Data Fig 2C).

Reviewer #2:

Remarks to the Author:

This study from the Serafini lab provides the first overview of ILC3 motility and tissue scanning in multiple anatomical areas of the intestine. The authors provide evidence for an immotile behavior of ILC3 under steady state conditions in immune competent mice. Upon inflammatory challenge, ILC3 become motile and initiate tissue scanning behavior.

In the absence of T cells, ILC3 motility and tissue scanning at steady state are increased, driven by CXCL25 and T cell-expressed CCR9, suggesting that T cells compete with ILC3 for CCL25 under steady state conditions.

The manuscript addresses an important knowledge gap in our understanding of ILC3 tissue behavior, defining the motility patterns of ILC3 during homeostasis and after inflammation. On the other hand, and perhaps due to the limitations of the Letter format, it remains to be determined what the impact of these behaviors is on ILC3 function and intestinal physiology.

Main points:

The rationale for investigating only CCL25 after combined neutralization of 4 chemokines is not convincing and does in no way exclude important functions of other chemokines. The authors should at least show that combined neutralization of the remaining 3 chemokines (CXCL12, CXCL16 and CCL21) is inferior to CCL25 in inhibiting ILC3 motility.

This study shows that increased ILC3 motility is a feature linked to inflammation in immune competent mice. However, studies into the mechanisms of ILC3 motility are only conducted under steady state conditions in the absence of T cells. As motility in response to inflammation is arguably the more physiological relevant condition, the role of CCL25 in inflammation-induced ILC3 motility should be addressed.

The hypothesis of competition for CCL25 by T cells and ILC3 is constructed from experiments in which CCR9^{+/+} T cells inhibit ILC3 motility while CCR9^{-/-} T cells do not. An important control in these experiments is to show the presence of normal numbers of CCR9^{-/-} T cells at the correct location in the intestinal villi. If T cells lacking CCR9 have a reduced ability to enter the intestine or do not properly locate to the villi, the mere reduction in T cells caused by these alterations, rather than the absence of competition for CCL25, would explain the observed ILC3 phenotype.

Other points:

Systemic treatment with flagellin does not equal “local environmental signals” (discussion). While it is clear from the data in the manuscript that inflammation leads to changes in ILC3 motility, the authors should be careful to draw conclusions on the mechanisms underlying this. From the current experiments it cannot be deduced whether ILC3 motility results from direct activation of ILC3 by flagellin, or from activation of other cells in the intestine (T cells?) or even cytokine alterations due to a heightened state of systemic inflammation. Understanding how ILC3 switch from immobile to mobile would add to the impact of the study, but might go beyond the letter format.

In Video S10 (CCL25 neutralization) it seems as if the numbers of GFP-only expressing cells decline with time after CCL25 neutralization. Could the authors comment on whether this is indeed the case?

Author Rebuttal to Initial comments
--

MANUSCRIPT # NI-LE32849

Jarade *et al.* Inflammation triggers ILC3 patrolling of the intestinal barrier

We thank the reviewers for their constructive comments on our manuscript and appreciate their enthusiasm for the novelty and quality of this work. After carefully reading the referees' comments, we have now addressed all their concerns, both major and minor. We believe that with the additional experiments, analyses and text, the study is substantially improved and appropriate for publication in *Nature Immunology*. Below, we have enclosed a detailed point-by-point response to the specific comments raised by the referees.

Reviewer #1 - Remarks to the Author:

The manuscript by Jarade et al., entitled "Inflammation triggers ILC3 patrolling of the intestinal barrier" seeks to investigate the behaviour of ILC3 populations within the small intestine using refined in vivo imaging approaches. After 10+ years of experiments revealing the requirement for ILC3 in intestinal homeostasis and defence, their actual behaviour within this tissue, which must contribute to how their roles are fulfilled, remains very poorly understood. Thus, I would concur with the authors that this is a key knowledge gap and to my knowledge this is the first study of its kind to really tackle this convincingly and enhance understanding of ILC migratory behaviour within the intestine. I think the findings are of clear interest to both mucosal immunologists but also more broadly across the immunology spectrum given the intense focus on ILC functions following their relatively recent discovery.

The manuscript is concise and clear in its basic discoveries, describing differential ILC3 behaviour associated with the two major ILC3 subsets present within the SILP (NCR+ and LTI-like/CCR6+). While the location of these populations was indicated by previous studies, a key initial discovery of this manuscript is that ILC3 populations are surprisingly static at 'steady state'. The authors further show that expression of IL-22 is largely restricted to the LTI-like population located within cryptopatches. To assess the migratory behaviour of ILC3, the authors employ the clever use of novel mice and bone marrow chimeras. This convincingly circumvents challenges in specific identification of ILC in the absence of T cell 'contamination' and overcomes the technical issues that I can envisage in definitively assessing ILC3 in vivo. The authors further provide evidence that acute inflammation drives altered behaviour within the villi ILC3 population inducing a patrolling phenotype which is clearly described. However, the relevance of i.v. flagellin versus local gut stimuli is a weak point in the study. The authors go on to show that the presence of an adaptive immune system impacts migration and provide reasonable evidence that availability of CCL25 controls the migratory ILC3 behaviour. Collectively these data provide a detailed and novel characterisation of the local migratory characteristics of ILC3 within the small intestine. In addition to some controls (expanded on below) to aid understanding of the experiments performed, where the manuscript is currently weaker is defining what the patrolling phenotype really means for the local intestinal environment and the extent to which the static ILC behaviour is tissue specific or a basic feature of these cells regardless of where they reside.

To develop the manuscript beyond a novel and interesting description, evidence of how inducing a patrolling behaviour benefits intestinal defence would build more functional relevance into the study. Given the extensive data linking ILC3 to regulation of intestinal T cell responses, the behaviour of the ILC3 within draining lymphoid tissue exploiting the clever imaging approaches used here would further advance the field.

We thank the reviewer for these supportive comments on our work.

Major comments

1. i.v. flagellin is used to induce a patrolling behaviour. While the data is clear, this systemic intervention should be contrasted with stimulations that better target the intestinal tract and provide more physiological insight into ILC3 behaviour and how this is altered. While not perfectly aligned to the small intestine, infection models such as Citrobacter or Salmonella would seem potentially appropriate to further investigate the behavioural responses of ILC in the small intestine in response to more local stimulation. Other options might be orally administered R848 + Ag.

As mentioned by the reviewer, an inflammatory/infection perturbation is a key component of our approach. We agree with the reviewer that an actual infection experiment would be interesting.

However, according to our institute's sanitary rules, mouse infection, using *C. rodentium* or *Salmonella*, requires containment in an A3 animal facility. Unfortunately, we do not have the possibility to perform 2-photon experiments in this area. Because of this technical limitation, we used a systemic inflammatory model (flagellin injection) which has several experimental advantages. First, we can easily induce a rapid and local IL-22 responses, in particular in the small intestine as shown in our experiment and as previously described by Van Maele *et al.*, J. Immunol 2010. Thus, a flagellin-induced inflammatory reaction can develop independently of any other antigen and is not associated with a possible tolerance mechanism, as observed in R848 + Ova model (Jumo *et al.* J. Immunol 2016 and Aumenier *et al.* Plos one 2010). Secondly, using this approach, ILC3 activation can be synchronized in time and space for all different intravital experiments, which is an important point in order to limit the number of mice and the experimental variability. Accordingly, we believe that our intravital imaging experiment using the flagellin model represents a relevant approach to understanding how a local and/or systemic inflammatory signal can modify ILC3 behavior and provides an important starting point for future studies in this area.

2. An obvious question raised by the altered behaviour of the ILC3 within the villi is what this means for the local environment. What changes within the villi when ILC3 patrol rather than remain static? What is the benefit here? Given that at steady state the authors provide evidence that IL-22 expression is restricted to the cryptopatch ILC3s, this suggests that IL-22 readily diffuses to support epithelial integrity in the gut. Is there data that IL-22 only acts very locally and thus patrolling helps dispense IL-22 at key locations? What then is the advantage of patrolling if IL-22 can diffuse? Perhaps other ILC3 functions are more relevant and IL-22 expression is a read-out but not the key function?

We thank the reviewer for making these points. We agree that the biological impact of patrolling ILC3s is a key issue. To assess the function of patrolling ILC3s on the intestinal environment, we blocked ILC3 patrolling behavior in *Rag2^{-/-}* mice by injection of anti-CCL25 (**Reviewer 1 - Fig. 1a**) and performed an analysis of intestinal permeability using oral FITC-dextran gavage. Our results show increased intestinal permeability in anti-CCL25-treated mice compared to isotype treated mice, suggesting a role of ILC3 patrolling in maintaining the intestinal barrier. We next attempted to identify the mechanism behind this protection. First, we tested whether this defect in intestinal permeability induced by anti-CCL25 antibodies might be related to altered expression of the intestinal barrier molecules (**Reviewer 1 - Fig. 1a**). Using a transcriptional analysis, we showed that expression of key components of the intestinal epithelium, such as tight junction proteins, antimicrobial peptides, and inflammatory cytokines were unchanged in the absence of patrolling ILC3s (**Reviewer 1 - Fig. 1b**). We then analyzed whether the lack of intestinal permeability could be associated with the death of epithelial cells. Using immunofluorescence staining of active-caspase 3, we observed an increase in intestinal epithelial cell (IEC) death in anti-CCL25 treated *Rag2^{-/-}* mice, suggesting that ILC3 patrolling behavior promotes epithelial integrity (**Reviewer 1 - Fig. 1c**). We next investigated whether this protection was mediated by IL-22 by assessing IEC death in *Rag2^{-/-} Il22^{-/-}* mice. We found that IEC death was significantly increase in absence of IL-22 compared to isotype and anti-CCL25 treated *Rag2^{-/-}* mice. However, further blockade of CCL25 did not major IEC death in *Rag2^{-/-} Il22^{-/-}* mice (**Reviewer 1 - Fig. 1c**). Thus, our results suggest that IL-22 acts to protect against early IEC death, and we hypothesize that patrolling ILC3s help to diffuse IL-22 within the villus (**Reviewer 1 - Fig. 1d**). We have now included these new results in the revised manuscript (**Figure 5**)

Reviewer 1 - Fig. 1: Impact of ILC3 motility on the intestinal barrier.

(a) Experimental design. *Rag2*^{-/-} mice were injected i.v. with either isotype control or anti-CCL25 blocking antibodies 18h and 4h before analysis. Then, mice were treated with FITC-dextran (12mg/g) by oral gavage and FITC dextran concentrations in blood sera were measured one hour later. Each bar corresponds to the mean ± SEM of the values obtained (***P*<0.01; two-tailed Mann-Whitney test) (b) Heatmap depicting tight junctions proteins, cytokines and antimicrobial peptides expression in ileal epithelial cells analyzed by Biomark (n=5). (c) Representative immunofluorescence of active-Caspase 3+ (Casp3; Red) staining in ileal intestinal epithelial cells (IEC) (scale bar 50µm). Graph shows absolute numbers of Act-Casp3⁺ IEC for each villus. Results are from two independent experiments (*Rag2*^{-/-}: n=4 and n=48 fields for isotype, n=4 and n=43 fields for anti-CCL25; *Rag2*^{-/-}*Il22*^{-/-}: n=3 and n=27 fields for isotype, n=3 and n=37 fields for anti-CCL25). Each bar corresponds to the mean ± SEM of the values obtained (***P*<0.01, ****P*<0.001; one-way ANOVA). (d) Schematic representation of patrolling ILC3 function on the intestinal barrier.

3. Can the authors provide evidence that the altered behaviour (i.e. patrolling in steady state) in *Rag*^{-/-} mice is not due to an altered microbiome and different stimuli feeding back into the system (i.e. rather than the absence of T cell down modulation of patrolling). It seems highly likely the microbiome is substantially altered in the absence of T cells. This potential confounder is not touched upon and some data to support that this is not microbiome driven would enhance the data.

The reviewer asks whether the microbiota could play a role in the generation of patrolling ILC3s in *Rag2*^{-/-} mice. In order to evaluate the impact of microbiota on ILC3 behavior, we treated *Rag2*^{-/-} mice for two weeks with a combination of antibiotics which strongly depletes microbiota in adult mice (Lochner *et al.* JEM 2011; **Reviewer 1 – Fig. 2a**). We found that ILC3s in antibiotic-treated *Rag2*^{-/-} *Rorc*^{GFP} *Il22*^{TdT} mice showed a reduction in velocity and an increase in arrest coefficient (**Reviewer 1 – Fig. 2b-d**), but not to the same extent as when T cells are present (**Fig. 3**). Moreover, ILC3 exploration of the intestinal tissue was not changed (**Reviewer 1 – Fig. 2b-d**). Thus, the microbiome appears to affect ILC3 motility but these microbiota changes do not fully recapitulate the effects of T cells on ILC3 motility. The microbiota may modify ILC3 patrolling, but the presence of T cells appears as the main suppressing signal (**Reviewer 1- Fig2. b-c**). Because antibiotic treatment does not mimic the effect of T cell reconstitution, it suggests that T cell suppression of ILC3 patrolling is not purely mediated through microbiota control. We have included these new results in the revised manuscript (**Extended Data Fig.4**).

Reviewer 1 - Fig. 2: Impact of the microbiota on ILC3 motility.

(a) Experimental design. *Rag2^{-/-}Rorc^{GFP}I122^{TdT}* (RR22) mice received antibiotic treatment (Abx; streptomycin 5g/L, ampicillin 1 g/L, colistin 1 g/L and sucrose 25 g/L) in drinking water. Two weeks later, intestine was imaged by multiphoton microscopy. (b-c) Individuals tracks of intestinal *I122⁻* and *I122⁺* ILC3s in RR22 mice, with or without Abx. (d) Graphs show mean speed, arrest coefficient and straightness ratio of indicated populations in RR22 and RR22 + Abx mice. Results are from one to three movies per condition from one of three independent experiments (n=257 and n=341 for *I122⁻* and *I122⁺* ILC3s in RR22 mice, respectively; and n=1124 *I122⁻* and 134 *I122⁺* ILC3s for RR22 + Abx mice, respectively). Each bar corresponds to the mean ± SEM of the values obtained (**P<0.001; one-way ANOVA).

4. The impact of the observations here would be enhanced by determining if ILC3 in other tissue locations are static. Appreciating that not all tissue compartments lend themselves to imaging, the mLN and ILC3 behaviour within the interfollicular zone would be an obvious place to look, given the specific location in this region (e.g. Mackley et al., 2015) and the link to MHCII-dependent regulation of T cell responses (e.g. Hepworth/Sonnenberg studies). To my knowledge, there is currently no data on the behaviour of ILC3 within the interfollicular region where they may migrate with a defined region or be more static and let activated T cells come to them. The authors have the tools to address this and the basic question of where ILC3 regulate responses (i.e. LN vs gut LP) remains unclear.

As mentioned by the reviewer, ILC3s play important roles in lymphoid organs for organogenesis and for protection against activated commensal bacteria-specific T cells. While we agree that a better understanding of ILC3 behavior in the LN would be interesting, new imaging approaches would be required to address this question. We respectfully submit that such technically challenging experiments should be considered as part of future studies.

Minor comments:

1. In Fig. 4b – is the CCR9 expression shown in WT or *Rag2^{-/-}* mice. Is CCR9 more abundantly expressed in *Rag2^{-/-}* ILC3?

We thank the reviewer for bringing up this point and we apologize for any confusion. In Fig. 4b, the CCR9 expression is shown in B6 mice which is now clearly indicated in the revised manuscript. As requested, we also tested the expression of CCR9 on ILC3 subsets in *Rag2^{-/-}* mice by flow cytometry (Reviewer 1 – Fig. 3a). We found that CCR9 expression on ILC3s is not modified in the absence of adaptive immune cells. This result supports the idea that the constitutive patrolling phenotype observed in *Rag2^{-/-}* mice is not due to modifications in chemokine receptor expression.

There is only some limited CCR9 expression shown on the NKp46 compartment. Does this indicate that only some ILC3 can patrol?

On the one hand, we cannot exclude that only NKp46⁺ ILC3s can patrol within intestinal villi. On the other hand, NKp46⁻ CD49a⁺ T-bet⁺ or CD49a⁻ T-bet⁻ CCR6⁻ ILC3 subsets are also present in the intestinal tissue. RNAseq analysis from the ImmGen core lab shows that NKp46⁻ CCR6⁻ ILC3s have low expression of *Ccr9* (Reviewer 1 - Fig. 3b) and our flow cytometry analysis of CCR9 expression on

ILC3 subsets confirms this finding (**Reviewer 1 - Fig. 3c**), suggesting that NKp46⁻ CD49a⁺ T-bet⁺ or CD49a⁻ T-bet⁻ CCR6⁻ ILC3s could also migrate within the intestinal tissue. However, our current work is limited by reporter mice available to image ILC3s. While the localization of these ILC3 subsets remains unknown, we believe that the analysis of their dynamic behavior can be the subject of future studies.

Reviewer 1 - Fig. 3: Analysis of CCR9 expression in WT and *Rag2*^{-/-} mice.

(a and c) Small intestine ILC3 subsets (NKp46⁺ ILC3s, CCR6⁺ ILC3s, CD49a⁺ NKp46⁻ ILC3s, CD49a⁻ CCR6⁻ ILC3s) and NK cells from B6 (a and c) and *Rag2*^{-/-} mice (a) were analyzed by flow cytometry. (b) Bar graph of *Ccr9* expression in thymic CD4⁺/CD8⁺ T cells and intestinal ILC3 subsets. Data were obtained from ImmGen (<http://rstats.immgen.org/Skyline/skyline.html>)

Do CCR9^{-/-} ILC3 still patrol?

Like the reviewer, we were interested in understanding the roles of CCR9 in ILC3 function. However, analyzing the impact of CCR9 ablation on ILC3 dynamics is limited as this receptor plays an essential role in gut homing of ILC3s. Kim *et al.*, Immunity 2015 previously described that ILC3 numbers and frequencies were drastically decreased in the intestinal villi of *Ccr9*^{-/-} mice. Moreover, the authors have shown that *Ccr9*^{-/-} ILC3s fail to normally populate the small intestine after bone marrow transplantation. Thus, analysis of *Ccr9*^{-/-} ILC3 dynamics (after adoptive transfer or in BM chimeras) in a normal ILC3 tissue compartment is not experimentally possible. As such, we decided to use a blocking antibody approach to bypass this issue and analyze the impact of the CCR9/CCL25 pathway on ILC3 dynamics.

2. In Fig. 3b, how well is the intestine reconstituted with T cells after transfer into the 'RR22' mice? What is the phenotype of the T cells here?

As already shown in **Fig. 3b** using intravital analysis of T cells, we observed that T cells (CFP⁺ cells) are present within intestinal villi following adoptive transfer, which are highly dynamic (**Extended Data Fig. 3c, Supplementary video 6**). As requested by the reviewer, we have analyzed the phenotype of T cells after cell transfer in RR22 mice (**Reviewer 1 – Fig. 4**). While all T cells have an activated phenotype (CD44⁺ CD62L⁻) as reported in previous published studies (Kim *et al.*, Frontiers in Immunol. 2018; Tomita *et al.*, J. Immunol 2008; Ge *et al.*, PNAS 2004), we found that CD8⁺ T cells, T_H17 cells (RORγt⁺ CD3⁺ CD5⁺ CD4⁺ cells), T_H2 cells (GATA3⁺ CD3⁺ CD5⁺ CD4⁺ cells) and regulatory T cell populations (Foxp3⁺ RORγt⁻ CD3⁺ CD5⁺ CD4⁺ cells) were present in the small intestine after T cell transfer to RR22 mice. We have included this information on the T cell phenotype in the revised manuscript (**Extended Data Fig. 3a**)

Reviewer 1 - Fig. 4: Analysis of T cell phenotype in *Rag2*^{-/-} + T cells mice.

Representative flow cytometry analysis of intestinal CD8a⁺ (a) and CD4⁺ (b) T cell subsets in *Rag2*^{-/-} mice two weeks post-adoptive transfer of T cells (data from one of three independent experiments).

There is some evidence from a limited number of mice (Ext Figure 5) that total T cell numbers are not significantly different. The data is somewhat limited. Is the phenotype of the T cells the same post-reconstitution? Given the role of CCR9 in SILP homing, is surprising that there is no impact.

As requested by the reviewer, we extended our analysis of T cells after *Ccr9*^{-/-} T cell transfer into *Rag2*^{-/-} mice. We found that the distribution of T cells was not affected by the absence of CCR9 (**Reviewer 1 – Fig. 5 a-c**). As previously observed in *Rag2*^{-/-} mice with WT T cells, all *Ccr9*^{-/-} T cells have an activated phenotype (CD44⁺CD62L⁻; **Reviewer 1 – Fig. 5a-b**). We did observe an impact of CCR9 on T_H17 cells, with a slight decrease in the frequencies of intestinal T_H17 cells compared to controls (**Reviewer 1 – Fig. 5b-d**), as previously reported by Wang C. *et al.*, J. Immunol. 2010. We have included the post-transfer analysis of *Ccr9*^{-/-} T cells in the revised manuscript (**Extended Data Fig. 7d-e**).

Reviewer 1 - Fig. 5: Analysis of T cell phenotype in *Rag2*^{-/-} + *Ccr9*^{-/-} T cells mice.

(a-b) Representative flow cytometry analysis of intestinal CD8a⁺ (a) and CD4⁺ (b) T cell subsets in *Rag2*^{-/-} mice two weeks post adoptive transfer of *Ccr9*^{-/-} T cells from one of two independent experiments. (c-d) Graphs show frequencies of total T cells (c), as well as CD4⁺ T cells (d), in the small intestine of *Rag2*^{-/-} mice two weeks post-adoptive transfer of either control *Ccr9*^{+/+} (n=4) or *Ccr9*^{-/-} (n=6) T cells. Each bar corresponds to the mean ± SEM of the values obtained (*P<0.05; two-tailed Mann-Whitney test)

3. In Fig. 2c is the total ILC compartment increase or are simply more NCR+ ILC3 turning on IL-22? This should be clarified – would be easy to show total numbers of the ILC3 subsets for example.

We thank the reviewer for this comment. As requested, we now provide additional quantification of absolute numbers of *Il22*^{(TdT)⁺} ILC3s as well as of IL-22⁺ ILC3s after flagellin administration. We found that absolute numbers of ILC3s are stable after flagellin injection (**Reviewer 1 – Fig. 6a**). Rather, our results show that IL-22 expression is increased following flagellin injection in all ILC3 subsets. NKp46⁺ ILC3s as well as CCR6⁻ CD49a⁻ and NKp46⁻ CD49a⁺ ILC3s become IL-22 positive at the RNA and protein levels, while CCR6⁺ ILC3s enhance IL-22 expression only at the protein level **Reviewer 1 – Fig. 6**). We have included these new results in the revised manuscript (**Extended Data Fig. 2d**).

Reviewer 1 - Fig. 6: Flagellin effect on ILC3 numbers and activation.

(a) Representative flow cytometry analysis of small intestine ILC3 subsets, pre-gated on CD45⁺ CD3⁻ CD5⁻ NK1.1⁻ CD127⁺ RORγt⁺ cells in WT mice from one of five experiments. Mice were treated with PBS or flagellin (5μg; *i.v.*) and *I/22*^{TdT} (transcript) and IL-22 (protein) expression in indicated ILC3 subsets were analyzed. (b) Graphs show absolute numbers of total ILC3s, *I/22*^{TdT} and IL-22⁺ ILC3s in PBS and flagellin treated mice, separated in ILC3 subsets. Each bar corresponds to the mean ± SEM of the values obtained (*P<0.05, **P<0.01, ***P<0.001; two-tailed Mann-Whitney test)

4. How does flagellin administration impact the cryptopatch ILC compartment? This isn't commented on even though the expression of IL-22 by the cryptopatch ILC3 (CCR6+) appears upregulated (Ext Data Fig 2C).

In isolated intestinal follicles, ILC3s are characterized by constitutive *I/22* expression at steady-state (Fig. 1b, Supplementary video 2), which is confirmed in our extended flow cytometry analysis where half of CCR6⁺ ILC3s constitutively express *I/22* (Reviewer 1 – Fig. 6). As mentioned in Reviewer 1 – Fig. 6b flagellin injection can rapidly induce expression of IL-22 at the protein level in CCR6⁺ ILC3s. Yet, we observed in various experiments that ILC3s in ILF remain immobile or migrate in very limited way compared to other cells following flagellin administration. For instance, in *Ncr1*^{GFP} *I/22*^{TdT} treated mice, GFP⁻ TdT⁺ cells remained clustered and still within ILF following flagellin treatment (Reviewer 1 – Fig. 7). GFP⁺ TdT⁻ cells (likely NK cells after flagellin treatment) are shown as a positive control of motility.

Reviewer 1 - Fig. 7: Flagellin effect on ILF ILC3 dynamics.

(a) Representative image (left, scale bar 20μm) of *Ncr1*^{GFP+} and *I/22*^{TdT+} cells in intestinal isolated lymphoid follicle of *Ncr1*^{GFP} *I/22*^{TdT} mice five hours after flagellin injection (5 μg). Individual tracks of GFP⁻TdT⁺ (yellow tracks) and GFP⁺TdT⁻ cells (green tracks) in intestinal follicles are shown.

Reviewer #2 - Remarks to the Author:

This study from the Serafini lab provides the first overview of ILC3 motility and tissue scanning in multiple anatomical areas of the intestine. The authors provide evidence for an immotile behavior of ILC3 under steady state conditions in immune competent mice. Upon inflammatory challenge, ILC3 become motile and initiate tissue scanning behavior.

In the absence of T cells, ILC3 motility and tissue scanning at steady state are increased, driven by CXCL25 and T cell-expressed CCR9, suggesting that T cells compete with ILC3 for CCL25 under steady state conditions.

The manuscript addresses an important knowledge gap in our understanding of ILC3 tissue behavior, defining the motility patterns of ILC3 during homeostasis and after inflammation. On the other hand, and perhaps due to the limitations of the Letter format, it remains to be determined what the impact of these behaviors is on ILC3 function and intestinal physiology.

We thank the reviewer for these supportive comments on our work.

Main points:

1. The rationale for investigating only CCL25 after combined neutralization of 4 chemokines is not convincing and does in no way exclude important functions of other chemokines. The authors should at least show that combined neutralization of the remaining 3 chemokines (CXCL12, CXCL16 and CCL21) is inferior to CCL25 in inhibiting ILC3 motility.

As requested by the reviewer, we analyzed the impact of the combined neutralization of CXCL12, CXCL16, and CCL21 on ILC3 motility. We found that ILC3 patrolling was only slightly modified 2h after injection of these three blocking antibodies (**Reviewer 2 – Fig. 1**), with a discrete reduction in speed mean (5.93 vs 4.92 $\mu\text{m}/\text{min}$) and increase in arrest coefficient (47.47 vs 55.34 %). Combined neutralization of these 3 chemokines on ILC3 patrolling is clearly inferior to CCL25 blockade in inhibiting ILC3 motility (**Fig. 5e**), confirming that CCL25 is a critical regulator of ILC3 patrolling. We have included these new results in the revised manuscript (**Extended Data Fig.6**).

Reviewer 2 - Fig 1: Impact of combined CXCL12, CXCL16 and CCL21 blockade on ILC3 motility.

(a) Experimental design. Intestinal ILC3s were imaged in *Rag2^{-/-} Rorc^{GFP} Il22^{TdT}* (RR22) mice using intravital imaging for one hour. Then, mice were injected *i.v.* with a combination of blocking monoclonal antibodies (anti-CXCL12, anti-CXCL16, and anti-CCL21; 50 μg each) and subsequently imaged for 2 hours. (b) Individual tracks of intestinal ILC3s before (-60-0 min; left) and 2h (60-120min; right) after blocking antibodies injection. (c) Graphs show mean speed, straightness ratio and arrest coefficient of intestinal ILC3s at indicated timepoints after blocking antibodies injection. Results in are from at least five movies per condition obtained in two independent experiments (n=129 and n=184 ILC3 tracks for 0h and 2h, respectively). Each bar corresponds to the mean \pm SEM of the values obtained (*P<0.05; **P<0.01; Kruskal-Wallis test).

2. This study shows that increased ILC3 motility is a feature linked to inflammation in immune competent mice. However, studies into the mechanisms of ILC3 motility are only conducted under steady state conditions in the absence of T cells. As motility in response to inflammation is arguably the more physiological relevant condition, the role of CCL25 in inflammation-induced ILC3 motility should be addressed.

To analyze the role of CCL25 in inflammation-induced ILC3 motility in immunocompetent mice, we first treated *Ncr1^{GFP} Il22^{TdT}* mice with flagellin to induce patrolling NKp46⁺ ILC3s in response to inflammation

and then injected anti-CCL25 blocking antibodies. As observed in *Rag2*^{-/-} mice, we found that inflammation-induced villus NKp46⁺ ILC3 patrolling is strongly affected following CCL25 neutralization in immunocompetent mice – as evidenced by a progressive decrease in ILC3 speed mean and an increase in arrest coefficient and confinement following CCL25 blockade (**Reviewer 2 – Fig. 2**). These results confirm the critical role of the CCR9/CCL25 axis for ILC3 motility within the gut during inflammation-induced ILC3 patrolling in immunocompetent mice and are now included in the revised manuscript (**Extended Data Fig.6**).

Reviewer 2 – Fig 2: Impact of CCL25 on inflammation-induced ILC3 motility in immunocompetent mice.

(a) Experimental design. Intestinal ILC3s were imaged in *Ncr1*^{FP} *I122*^{TdT} mice using intravital imaging. First, mice were *i.v.* injected with flagellin (5 µg) five hours prior intravital imaging. Then, mice were *i.v.* injected *i.v.* with CCL25 blocking antibody (anti-CCL25; 50 µg) and subsequently imaged for 2 hours. (b) Individual tracks of intestinal NKp46⁺ *I122*⁺ ILC3s before (-60-0 min; left) and 2h (60-120min; right) after anti-CCL25 injection. (c) Graphs show mean speed, arrest coefficient and straightness ratio of intestinal NKp46⁺ *I122*⁺ ILC3s at indicated timepoints after anti-CCL25 injection. Results are from at least five movies per condition obtained in two independent experiments (n=78 cells for 0h =50 for 1h and n=49 cells for 2h). Each bar corresponds to the mean ± SEM of the values obtained (**P<0.01; ***P<0.001; Kruskal-Wallis test).

3. The hypothesis of competition for CCL25 by T cells and ILC3 is constructed from experiments in which CCR9+/+ T cells inhibit ILC3 motility while CCR9-/- T cells do not. An important control in these experiments is to show the presence of normal numbers of CCR9-/- T cells at the correct location in the intestinal villi. If T cells lacking CCR9 have a reduced ability to enter the intestine or do not properly locate to the villi, the mere reduction in T cells caused by these alterations, rather than the absence of competition for CCL25, would explain the observed ILC3 phenotype.

We fully agree with the reviewer that the presence of normal numbers of *Ccr9*^{-/-} T cells in the intestinal villi represents one important feature to construct our hypothesis of the importance of CCR9⁺ T cells in ILC3 biology. In our original submission, we already showed that the number of *Ccr9*^{-/-} T cells in villi appears to be normal using flow cytometry (**Extended Data Fig. 7b**) and we confirmed their presence within intestinal villi through immunofluorescence staining on ilea of adoptively transferred mice (**Extended Data Fig. 7c**). This result did not seem to be clearly written and we corrected this point in the revised manuscript. To consolidate this finding, we have now extended our T cell analysis to include the phenotype of *Ccr9*^{-/-} T cells after cell transfer in *RR22* mice (**Reviewer 2 – Fig. 3** and **Extended Data Fig. 7d, e**). These new results are included in the revised manuscript.

Reviewer 2 - Fig. 3: Analysis of T cell phenotype in *Rag2*^{-/-} + *Ccr9*^{+/+} and *Ccr9*^{-/-} T cells mice.

(a-d) Representative flow cytometry analysis of intestinal CD8a⁺ and CD4⁺ T cell subsets in *Rag2*^{-/-} mice two weeks post adoptive transfer of either *Ccr9*^{+/+} (a-b) or *Ccr9*^{-/-} T cells (c-d) from one of two independent experiments. (e-f) Graphs show frequencies of total T cells (e), as well as CD4⁺ T cells (f), in the small intestine of *Rag2*^{-/-} mice two weeks post-adoptive transfer of either control *Ccr9*^{+/+} (n=4) or *Ccr9*^{-/-} (n=6) T cells. Each bar corresponds to the mean ± SEM of the values obtained (*P<0.05; two-tailed Mann-Whitney test).

Other points:

1. Systemic treatment with flagellin does not equal “local environmental signals” (discussion). While it is clear from the data in the manuscript that inflammation leads to changes in ILC3 motility, the authors should be careful to draw conclusions on the mechanisms underlying this. From the current experiments it cannot be deduced whether ILC3 motility results from direct activation of ILC3 by flagellin, or from activation of other cells in the intestine (T cells?) or even cytokine alterations due to a heightened state of systemic inflammation. Understanding how ILC3 switch from immobile to mobile would add to the impact of the study, but might go beyond the letter format.

The cellular and molecular pathways that activate ILC3s are diverse and varied. ILC3 motility is not likely to be induced by flagellin directly, as the *Tlr5* gene is not expressed in ILC3 subsets (Immgen database; **Reviewer 2 – Fig. 4**), a point we now include in the revised manuscript. Nevertheless, recent studies have shown that ILC3s are ideally positioned to sense a wide range of external and host-derived signals including factors from stromal cells, innate and adaptive immune cells, neurons, microbes, and metabolites (almost 15 pathways; reviewed in Zhou W. and Sonnenberg GF Trends Immunol. 2020), which may or may not be involved in the initiation of ILC3 patrolling. The analysis of these pathways requires new tools and the development of new imaging approaches to address this question. We respectfully submit that such technically challenging experiments could be considered in the future.

Reviewer 2 - Fig. 4: Tlr5 expression in ILC3 subsets

Bar graph of Tlr5 expression in bone marrow neutrophils and intestinal ILC3 subsets. Data are obtained from ImmGen (<http://rstats.immgen.org/Skyline/skyline.html>).

2. In Video S10 (CCL25 neutralization) it seems as if the numbers of GFP-only expressing cells decline with time after CCL25 neutralization. Could the authors comment on whether this is indeed the case?

As requested by the reviewer, we analyzed the numbers of GFP⁺ ILC3s per villus before and after CCL25 neutralization. We found that the number of GFP⁺ ILC3s was not affected by the presence of CCL25 blocking antibody. (**Reviewer 2 – Fig. 5**).

Reviewer 2 - Fig. 5: Effect of CCL25 neutralization on ILC3 numbers over time.

Graph shows the number of GFP⁺ ILC3s at indicated timepoints after anti-CCL25 blocking antibody injection. Each bar corresponds to the mean \pm SEM of the values obtained (one-way ANOVA).

Decision Letter, first revision:

24th May 2022

Dear Dr Serafini,

Thank you for your response to the reviewers' comments on your article, "Inflammation triggers ILC3 patrolling of the intestinal barrier". Although we are interested in the possibility of publishing your study in Nature Immunology, the issues raised by the referees need to be addressed.

Regarding the model used, systemic injection of flagellin does not represent local inflammation, and it cannot be excluded that the effects seen are dependent on activation of immune cells in other peripheral site. As such, please revise to refer to the model as a general model of inflammation, which is a more accurate description of the results in the study. Please remove the textual references to "local" intestinal inflammation. Regarding the other issues raised by the referees, please revise along the lines specified in your letter and to address all the remaining concerns from the referees. In addition, please revise keeping in mind the formatting requirements for a Letter (5 main display items, 2500 words for the main text and 30 references).

At resubmission, please include a "Response to referees" detailing, point-by-point, how you addressed each referee comment. If no action was taken to address a point, you must provide a compelling argument. This response will be sent back to the referees along with the revised manuscript. Reporting summary:

We hope to receive your revised manuscript within three-four weeks. If you cannot send it within this time, please let us know. We will be happy to consider your revision so long as novelty has not been compromised in the interim.

Nature Immunology is committed to improving transparency in authorship. As part of our efforts in this direction, we are now requesting that all authors identified as 'corresponding author' on published papers create and link their Open Researcher and Contributor Identifier (ORCID) with their account on the Manuscript Tracking System (MTS), prior to acceptance. ORCID helps the scientific community achieve unambiguous attribution of all scholarly contributions. You can create and link your ORCID from the home page of the MTS by clicking on 'Modify my Springer Nature account'. For more information please visit www.springernature.com/orcid.

Sincerely,

Ioana Visan, Ph.D.
Senior Editor
Nature Immunology

Tel: 212-726-9207
Fax: 212-696-9752
www.nature.com/ni

Reviewers' Comments:

Reviewer #1:

Remarks to the Author:

In the study of Jarade et al., the authors explore the dynamic behaviour of ILC3 within the intestine, revealing subset specific differences in key function (IL-22+ expression) as well as behaviour – the CCR6 compartment remains static while the NCR+ ILC3 population can be induced to patrol. As noted in the original review these observations of the dynamic behaviour of ILC3 within the intestinal tissue are novel and of interest. The mechanisms impacting this patrolling action are explored and reveal CCL25 as a contributing factor with competition with T cells for this chemokine a further aspect. To my mind the real strength of this manuscript is that it starts to fill the void in our understanding of how ILC3 behave in vivo.

The key limitations of the initial version of the study to my mind were

i) the relevance of the inflammatory model used to induce the patrolling behaviour – this was systemic flagellin administration which was quite a conceptual stretch to local cues within the gut

ii) the relevance of this behaviour to intestinal defence

iii) the extent to which this is specific to the ILC3 compartment of the intestine, particularly the 'static' CCR6+ compartment that dominates the gut associated lymphoid tissue and is thought to regulate T cell responses through as yet poorly defined mechanisms.

Alongside this I raised concerns regarding an altered microbiome in Rag-/- mice (major concern) and more minor comments regarding controls and clarifying specific details.

Regarding the major concerns, the authors:

1. argue that their systemic flagellin model is appropriate to understanding 'how a local and/or systemic inflammatory signal can modify ILC3 behavior'. While I can accept that the approach does model a systemic inflammatory signal, it clearly does not inform of a local signal. The authors state that they are unable to perform infection studies in combination with the imaging platform due to conditions within their facility – I can understand this given rules regarding where infection studies and infected mice can be within facilities. It is disappointing that the authors essentially refused to consider an alternative approach that could have tried to model a more intestinal focused inflammatory signal. The link with R848 and potential tolerance models is not relevant and something of a red-herring. Through not developing the inflammatory approach beyond the sledgehammer effect of systemic flagellin, I think the

potential relevance of the findings remains more limited. I struggle to see that there were not ways to investigate this further had the researchers been of a mind to do this.

2. Present some evidence that in the absence of patrolling ILC3 in Rag^{-/-} mice (with anti-CCL25) this modestly increases IEC death, which the authors interpret as evidence that ILC3 patrolling facilitates IL-22 diffusion. While these observations do comprise some further experimental data, an obvious resulting question is how this fits with the evidence that under homeostatic/non-inflammatory conditions, IL-22 diffusion from the crypt (where IL22+ CCR6+ ILC3 are located) is sufficient? In short, the new data does not completely address the basic question of what is the point of the ILC3 patrolling behaviour?

3. the authors decline to investigate ILC3 behaviour at other sites citing this to be technically challenging and beyond the scope of the manuscript. As initially stated I think this would have broadened and strengthened the manuscript.

4. The authors have assessed the impact of the microbiota through Abx treatment and observed effects on ILC3 speed and arrest coefficient, but, as the authors conclude, this is modest compared to the effect of the presence of T cells and thus it doesn't appear that an altered microbiome is a key driver of the patrolling function.

Regarding the minor concerns:

The author responses essentially address the concerns raised. Data provided in response to Q2 (T cell reconstitution) is somewhat limited in phenotyping but does provide better information regarding the reconstitution. It is also clear that the composition in these mice is quite distinct from a WT setting. I think it is appropriate for the authors to insert a line of text in regard to Extended Data Fig 3 making clear that they are not claiming that a 'normal' T cell compartment is now established, rather, their data does indicate that the presence of T cells within the intestine alters the behaviour of the exploratory ILC3.

In conclusion, the authors have adequately dealt with the minor comments while providing some new experimental data in response to the major concerns. As a result of this the manuscript is only modestly changed by the review process and I think the strengths and weaknesses of the study largely remain. The manuscript does provide novel insight into ILC3 behaviour in the intestine that I think will be of broad interest to the immunology field. The authors further explore some of the mechanisms regulating this and provide evidence for a role for CCL25. The weaknesses of the study are still centred around what this actually means. Since the patrolling behaviour has only been induced and studied in a highly artificial manner, it is unclear what this really means for intestinal ILC3 responses to local insult. Mechanistic insight into the impact of patrolling remains poorly defined.

Reviewer #2:

Remarks to the Author:

With the revised version of the manuscript, the authors have addressed all my comments, both experimentally and in writing.

One additional remark. In extended data figure 3B, describing R22 + T cells + flagellin, the micrograph is a selected area from Figure 2B, describing R22 mice that did not receive flagellin. Although it will not affect the data, it will be more appropriate to replace this micrograph with an image from a flagellin experiment.

Author Rebuttal, first revision:

Reviewer #1

(Remarks to the Author)

In the study of Jarade et al., the authors explore the dynamic behaviour of ILC3 within the intestine, revealing subset specific differences in key function (IL-22+ expression) as well as behaviour – the CCR6 compartment remains static while the NCR+ ILC3 population can be induced to patrol. As noted in the original review these observations of the dynamic behaviour of ILC3 within the intestinal tissue are novel and of interest. The mechanisms impacting this patrolling action are explored and reveal CCL25 as a contributing factor with competition with T cells for this chemokine a further aspect. To my mind the real strength of this manuscript is that it starts to fill the void in our understanding of how ILC3 behave in vivo. The key limitations of the initial version of the study to my mind were:

i) the relevance of the inflammatory model used to induce the patrolling behaviour – this was systemic flagellin administration which was quite a conceptual stretch to local cues within the gut.

ii) the relevance of this behaviour to intestinal defence.

iii) the extent to which this is specific to the ILC3 compartment of the intestine, particularly the ‘static’ CCR6+ compartment that dominates the gut associated lymphoid tissue and is thought to regulate T cell responses through as yet poorly defined mechanisms.

Alongside this I raised concerns regarding an altered microbiome in Rag-/- mice (major concern) and more minor comments regarding controls and clarifying specific details.

We thank the reviewer for his/her comments on our manuscript. All major and minor points raised during the initial round of review (and summarized above by the reviewer) were fully and carefully considered and we addressed all points by additional experimentation and/or rebuttal during resubmission. We thank the reviewer for these comments which provoked additional experimentation resulting in new information that improved our study.

Regarding the major concerns, the authors:

1. argue that their systemic flagellin model is appropriate to understanding ‘how a local and/or systemic inflammatory signal can modify ILC3 behavior’. While I can accept that the approach does model a systemic inflammatory signal, it clearly does not inform of a local signal. The authors state that they are unable to perform infection studies in combination with the imaging platform due to conditions within their facility – I can understand this given rules regarding where infection studies and infected mice can be within facilities. It is disappointing that the authors essentially refused to consider an alternative approach that could have tried to model a more intestinal focused inflammatory signal. The link with R848 and potential tolerance models is not relevant and something of a red-herring. Through not developing the inflammatory approach beyond the sledgehammer effect of systemic flagellin, I think the potential relevance of the findings remains more limited. I struggle to see that there were not ways to investigate this further had the researchers been of a mind to do this.

We used the intravenous flagellin model as it has been clearly shown to provoke a local inflammatory effect in the intestine. To completely dismiss this approach as a means to interrogate local mucosal inflammation is not reasonable as it ignores an extensive published literature on this model. TLR5 is strongly expressed on the basolateral side of intestinal epithelial cells and by mononuclear phagocytes (reviewed in Immgen database and PMID: 16497588). Accordingly, TLR5 is used by the mucosal immune system to detect flagellin in the context of microbiota surveillance. Aberrantly elevated TLR5 activation or absence of TLR5 expression results in an impairment of the intestinal barrier and gut microbial composition (PMID: 28146004, 25172014). Previous studies have shown that flagellin injection promotes a transient intestinal T_H17-related response (PMID:20566828, 22306017) and elicits mucosal antibodies (PMID: 31827095). E. Pamer and colleagues have shown that systemic administration of flagellin induces epithelial expression of antimicrobial peptides (AMPs), which is IL-22-dependent (PMID: 22306017). Using conditional depletion of dendritic cell (DC) subsets, they demonstrated that specific CD103⁺CD11c⁺ cells, but not monocyte-derived DC, are necessary to activate IL-22 dependent AMP production. Therefore, when flagellin is delivered via the systemic route, the activation of the IL-22 response also occurs at the local intestinal level. Accordingly, this approach allows for strong, fast, and synchronized activation of gut ILC3s, which are essential points required to identify new concepts in the intestinal ILC3 biology. As our aim was to provide a first description of

intestinal ILC3 dynamics following an inflammatory reaction, the flagellin model used appears reasonable and pertinent.

We would like to point out that the initiation of specific local activation remains an important and challenging question for many studies on intestinal biology. Nevertheless, we are not alone in using systemic triggers to study local inflammatory processes. Recently, G. Sonnenberg and colleagues used a similar approach to define the role of ILC3 in TNF-mediated gut inflammation (PMID: 35102343). In their work, they injected TNF- α systemically to analyze local inflammatory reactions in the intestine. Although TNF- α has the potential to affect the entire body, they concluded that intestinal ILC3s have a critical role in preventing TNF-mediated inflammation in the gut.

In the first round of revision, the reviewer proposed several alternative models (including gut infection and oral 'R848 + Ag') to access 'local' inflammation in the intestine. As we rebutted in our previous resubmission, intravital imaging of infected animals is not possible at our institute, and this response was accepted by the Reviewer #1. Concerning the proposed oral 'R848 + Ag' model, upon closer inspection, this approach would appear to have several limitations. First, intestinal ILC3 activation has never been demonstrated in this model. Second, the potential activation of ILC3 in the 'R848 + Ag' model would be by definition antigen-dependent (implicating T cell activation) which is an issue given the inter-dependence of ILC3 and T cell activation. Third, the activation of IL-22-producing ILC3 is dependent on the CD11c⁺ population and their capacity to produce IL-23 (PMID: 25024136). However, the administration of oral R848 relocates intestinal dendritic cells to the draining lymph node within 2 to 12h (PMID: 16621985). Finally, several studies reported systemic cytokine effects of oral R848 gavage in both rodent and human models (PMID: 16621985, 17532523). Together, by changing several important aspects of intestinal homeostasis, the proposed oral R848 model would be difficult to interpret and would also be subject to the same criticism as flagellin administration due to its known systemic effects.

Overall, we would argue that our flagellin approach remains an appropriate model to interrogate local inflammatory effects on intestinal ILC3. Still, as systemic effects have been documented during flagellin administration, we cannot formally exclude that some effects may be dependent on activation of immune cells in other peripheral sites. We have therefore removed textual references to 'local inflammation' and modified the following statement in the revised manuscript:

"Ncr1^{GFP}IL22^{TdT} mice were injected with bacterial flagellin that can indirectly activate ILC3s, via TLR5⁺ myeloid cell stimulation, thereby mimic bacterial infection-induced inflammation^{14,15}. [...] Together, these data indicate that environmental signals, such as acute generalized inflammation caused by systemic bacterial flagellin administration, impact ILC3 behavior and induce a patrolling function that is associated with enhanced IL-22 expression". (Revised manuscript, lines 10-13 [...] 21-23, page 3)

2. Present some evidence that in the absence of patrolling ILC3 in Rag-/- mice (with anti-CCL25) this modestly increases IEC death, which the authors interpret as evidence that ILC3 patrolling facilitates IL-22 diffusion. While these observations do comprise some further experimental data, an obvious resulting question is how this fits with the evidence that under homeostatic/non-inflammatory conditions, IL-22 diffusion from the crypt (where IL22+ CCR6+ ILC3 are located) is sufficient? In short, the new data does not completely address the basic question of what is the point of the ILC3 patrolling behaviour?

We would respectfully submit that our novel observations are not "modest" but instead strongly supported by two independent lines of evidence based on in vivo analyses.

In the first round of review, an interesting question was raised by this reviewer concerning the biological impact of patrolling ILC3s: "An obvious question raised by the altered behavior of the ILC3 within the villus what this means for the local environment? What changes within the villus when ILC3 patrol rather than remain static? What is the benefit here?" (quotations taken from the initial round of review). The reviewer requested experiments to further examine whether the intestinal villus was affected by the restriction of patrolling ILC3s. We performed three experiments to address this concern by analyzing the global impact on intestinal barrier functions and underlying cellular mechanisms. These new results have been included in the revised manuscript and clearly provide important new information regarding the ILC3 patrolling response as requested by the reviewer.

As observed by several groups, the role of IL-22 in villi and crypts appears functionally different. We have previously shown that ablation of CXCR6 generated a selective loss of villus NKp46⁺ ILC3. While CCR6⁺ ILC3s were not affected, IL-22 produced in the crypt was not sufficient to protect the host against bacterial infection (PMID:24456160). These observations opened the possibility for different roles for IL-22 present in villi and crypts. Here, our data reinforce this hypothesis and identify specific features of compartmentalized ILC3 under steady-state and inflammatory conditions. Nevertheless, the role of ILC3-derived IL-22 in the crypt is only partially understood. As a strong expression of IL-22 in CCR6⁺ ILC3 is observed, we and others suspect that the high concentration of IL-22 in this area is sufficient to have a local effect. As observed by T. Cupedo and colleagues (PMID:26392223), ILC3s preserve intestinal stem cells after inflammation. IL-22 blockade led to significant loss of intestinal stem cell maintenance in the duodenum. However, stem cells in jejunum and ileum were not affected, indicating that IL-22 is one, not the only factor involved in crypt homeostasis. Overall, the functional role of CCR6⁺ IL-22^{+/-} ILC3 remains to be clarified, but our results provide a useful starting point for future studies on these specific cells.

In summary, we strived to use the best possible models to gain a mechanistic understanding of signals that influence ILC3 behavior. We believe that our revised manuscript provides all the necessary evidence to fully justify the conclusions that patrolling ILC3s promote mucosal immunosurveillance and support barrier homeostasis.

3. the authors decline to investigate ILC3 behaviour at other sites citing this to be technically challenging and beyond the scope of the manuscript. As initially stated I think this would have broadened and strengthened the manuscript.

In the first round of review, a question was raised concerning the ILC3 migration in other tissue locations. As observed by many groups, ILC3s are present in lymphoid tissue, mucosal tissue (lung, gut, skin), and liver. It remains unclear what ILC3s do and where they are located in most of these organs. We focused our attention on the gut where the majority of ILC3 are found. As such, our study provides the first in-depth intravital analysis of intestinal ILC3 dynamics and behavior. As acknowledged by Reviewer #1, our work fills a “key knowledge gap” in this field.

Based on our experience, we know that intravital analysis requires specific tools to properly describe cell dynamics and behavior. The reviewer underlines the crucial role of ILC3 in fetal lymphoid organogenesis and the protection against activated commensal bacteria-specific T cells in adult animals. We agree that a better understanding of ILC3 behavior in the lymph node might help to address these (and other) interesting questions. Still, due to the very low frequency of NKp46⁺ ILC3 in lymph nodes and the developmental dependency of adaptive immune cells for lymph node (LN) development, we cannot use Ncr1^{GFP} I122^{TdT}, Rag^{-/-}Rorc^{GFP}I122^{TdT}, or chimera models to perform these studies. We respectfully submit that developing specific ILC3 CCR6 reporters (which are not yet available) may be necessary and that these experiments should be the subject of future studies.

Overall, we respectfully stand by our rebuttal that the analysis of LN ILC3 is not a prerequisite to understand the behavior of intestinal ILC3. The novel observations already included in our manuscript are not trivial but to quote Reviewer #1: ‘*this is the first study of its kind to really tackle this convincingly and enhance understanding of ILC migratory behaviour within the intestine*’.

4. The authors have assessed the impact of the microbiota through Abx treatment and observed effects on ILC3 speed and arrest coefficient, but, as the authors conclude, this is modest compared to the effect of the presence of T cells and thus it doesn't appear that an altered microbiome is a key driver of the patrolling function.

In the first round of review, the reviewer requested an analysis of how microbiota influenced ILC3 patrolling (Major point #4). We performed experiments as requested and found that antibiotic treatment (that significantly reduced commensal microbiota) was not sufficient to modify ILC3 behavior. These new experiments were included in the revised manuscript and clearly exclude an altered microbiome as a major regulator of ILC behavior.

Regarding the minor concerns:

The author responses essentially address the concerns raised. Data provided in response to Q2 (T cell reconstitution) is somewhat limited in phenotyping but does provide better information regarding the reconstitution. It is also clear that the composition in these mice is quite distinct from a WT setting. I think it is appropriate for the authors to insert a line of text in regard to Extended Data Fig 3 making clear that they are not claiming that a 'normal' T cell compartment is now established, rather, their data does indicate that the presence of T cells within the intestine alters the behaviour of the exploratory ILC3.

This point has been addressed in the revised manuscript:

"Thus, the presence of T cells appears to suppress ILC3 patrolling behavior under steady-state conditions." (Revised manuscript, page 3)

In conclusion, the authors have adequately dealt with the minor comments while providing some new experimental data in response to the major concerns. As a result of this the manuscript is only modestly changed by the review process and I think the strengths and weaknesses of the study largely remain. The manuscript does provide novel insight into ILC3 behaviour in the intestine that I think will be of broad interest to the immunology field. The authors further explore some of the mechanisms regulating this and provide evidence for a role for CCL25. The weaknesses of the study are still centred around what this actually means. Since the patrolling behaviour has only been induced and studied in a highly artificial manner, it is unclear what this really means for intestinal ILC3 responses to local insult. Mechanistic insight into the impact of patrolling remains poorly defined.

We are very surprised by these comments from Reviewer #1. In the initial round of review, we appreciated his/her enthusiastic comments, and the different critiques and suggestions helped clarify important aspects of our work and strengthened the overall message. The reviewer raised 10 points (4 major, 6 minor) concerning the role of patrolling ILC3, the impact of microbiota, the expression of CCR9, the use of different models, reconstitution of T cells, and ILC3 phenotypes (villus and crypt). All of these points were addressed: 9 new experiments were performed (involving 83 mice) and results included in the revised manuscript. Based on these new results, it would be unfair to claim that our revised manuscript has only been modestly modified.

While the biological roles for ILC3 in intestinal homeostasis and immunity are partially known, the spatiotemporal regulatory mechanisms that control ILC3 responses and their cellular dynamics within tissues are undetermined. Here, we provide the first description of intestinal ILC3 dynamics using intravital live imaging in steady-state, inflammation, and in the absence of adaptive immune cells, using 5 different models. We further show that ILC3s exhibit an exclusive patrolling phenotype depending on 3 key regulators. Finally, we delineate a new function of ILC3s in the intestinal barrier maintenance, using histological analysis of IEC death. Although our models can be considered artificial, we believe our approach uses an appropriate way to analyze ILC3 behavior and contributes to a better understanding of the complex role that ILC3s play in the intestine.

Reviewer #2

(Remarks to the Author)

With the revised version of the manuscript, the authors have addressed all my comments, both experimentally and in writing.

We thank the reviewer for constructive comments on our paper.

One additional remark. In extended data figure 3B, describing R22 + T cells + flagellin, the micrograph is a selected area from Figure 2B, describing R22 mice that did not receive flagellin. Although it will not affect the data, it will be more appropriate to replace this micrograph with an image from a flagellin experiment.

This point has been addressed in the revised manuscript (Revised version - Extended Data Fig.3b)

Final Decision Letter:

In reply please quote: NI-LE32849C

Dear Dr. Serafini,

I am delighted to accept your manuscript entitled "Inflammation triggers ILC3 patrolling of the intestinal barrier" for publication in an upcoming issue of Nature Immunology.

Over the next few weeks, your paper will be copyedited to ensure that it conforms to Nature Immunology style. Once your paper is typeset, you will receive an email with a link to choose the appropriate publishing options for your paper and our Author Services team will be in touch regarding any additional information that may be required.

Please note that *Nature Immunology* is a Transformative Journal (TJ). Authors may publish their research with us through the traditional subscription access route or make their paper immediately open access through payment of an article-processing charge (APC). Authors will not be required to make a final decision about access to their article until it has been accepted. [Find out more about Transformative Journals](https://www.springernature.com/gp/open-research/transformative-journals).

Your paper will be published online soon after we receive your corrections and will appear in print in the next available issue. Content is published online weekly on Mondays and Thursdays, and the embargo is set at 16:00 London time (GMT)/11:00 am US Eastern time (EST) on the day of publication. Now is the time to inform your Public Relations or Press Office about your paper, as they might be interested in promoting its publication. This will allow them time to prepare an accurate and satisfactory press release. Include your manuscript tracking number (NI-LE32849C) and the name of the journal, which they will need when they contact our office.

About one week before your paper is published online, we shall be distributing a press release to news organizations worldwide, which may very well include details of your work. We are happy for your institution or funding agency to prepare its own press release, but it must mention the embargo date and Nature Immunology. Our Press Office will contact you closer to the time of publication, but if you or your Press Office have any enquiries in the meantime, please contact press@nature.com.

Also, if you have any spectacular or outstanding figures or graphics associated with your manuscript - though not necessarily included with your submission - we'd be delighted to consider them as candidates for our cover. Simply send an electronic version (accompanied by a hard copy) to us with a possible cover caption enclosed.

Please note that we encourage the authors to self-archive their manuscript (the accepted version before copy editing) in their institutional repository, and in their funders' archives, six months after publication. Nature Portfolio recognizes the efforts of funding bodies to increase access of the research they fund, and strongly encourages authors to participate in such efforts. For information about our editorial policy, including license agreement and author copyright, please visit www.nature.com/ni/about/ed_policies/index.html

Sincerely,

Ioana Visan, Ph.D.
Senior Editor
Nature Immunology

Tel: 212-726-9207

Fax: 212-696-9752
www.nature.com/ni